# SDUST2021GRA: Global marine gravity anomaly model recovered from Ka-band and Ku-band satellite altimeter data

Chengcheng Zhu[1,2], Jinyun Guo[1], Jiajia Yuan[1,3], Zhen Li[1], Xin Liu[1], Jinyao Gao[4]

[1]College of Geodesy and Geomatics, Shandong University of Science and Technology, Qingdao, Shandong, China
[2]School of Surveying and Geo-Informatics, Shandong Jianzhu University, Jinan, Shandong, China
[3]School of Geomatics, Anhui University of Science and Technology, Huainan, Anhui, China
[4]Second Institute of Oceanography of MNR, Hangzhou, Zhejiang, China

*Correspondence to*: Jinyun Guo (jinyunguo1@126.com)

**Abstract.** With the launch of altimetry satellites with different observation frequencies and different survey missions, it is
necessary to integrate multi-satellites altimeter data to establish a new global marine gravity anomaly model. Based on Ka-band SSHs from SARAL/AltiKA and Ku-band SSHs from other satellites (including HY-2A) in geodetic missions and exact repeat missions, the global marine gravity anomaly model of SDUST2021GRA on a 1′×1′ grid is derived. Gridded deflections of vertical (DOVs) are determined from along-track geoid gradients by the least-squares collocation method, in which the noise variances of along-track geoid gradients are obtained by the iteration method for Ka-band geodetic mission
and by the SSH crossover discrepancies for other altimetry missions. SDUST2021GRA is recovered from the gridded DOVs by the inverse Vening-Meinesz formula, and analyzed by comparing with the recognized marine gravity anomaly models of DTU17 and SIO V30.1. Final, the accuracy of SDUST2021GRA, DTU17 and SIO V30.1 is assessed by preprocessed shipborne gravity anomalies. In conclusion, the differences between SDUST2021GRA and recognized models are small, indicating the reliability of SDUST2021GRA. The differences are mainly concentrated between -5 mGal and 5 mGal, which
accounts for more than 95% of the total number. Assessed by shipborne gravity, the accuracy of SDUST2021GRA is 2.37 mGal in the global, which is higher than that of DTU17 (2.74 mGal) and SIO V30.1(2.69 mGal). The precision advantage of SDUST2021GRA is mainly concentrated in offshore areas. HY-2A-measured altimeter data have an important role on gravity anomaly recovery in areas with complex coastlines and many islands. SDUST2021GRA is concluded to reach an international advanced level for the altimeter-derived marine gravity model, especially in the offshore area. The
SDUST2021GRA model are freely available at the site of https://doi.org/10.5281/zenodo.6668159 (Zhu et al., 2022).

## 1 Introduction

Accurate marine gravity anomalies play an important role in the fields of submarine topography (Sun et al., 2021), oceanic lithosphere (Kim and Wessel, 2011; Shahraki et al., 2018; Gozzard et al., 2019), Earth structure (Ebbing et al., 2018) and submarine exploitation (Sun et al., 2018). The technique of satellite altimetry is widely applied to construct local and global
marine gravity anomaly models (Andersen and Knudsen, 2019; Zhu et al., 2020; Sandwell et al., 2021; Guo et al., 2022).

With the launch of different altimetry satellites, a large number of altimeter data have been obtained. With the performance of the geodetic mission (GM) of altimeter satellite, the density of altimeter data can meet the requirements of inversion of high-resolution and high-precision gravity anomaly models, e.g., CryoSat-2 provided a nominal track spacing of less than 2.5 km (Sandwell et al., 2014b; Ji et al., 2021b) after about 10 years in orbit. Meanwhile, different observation techniques are used in different altimetry satellites, e.g., the Ka-band altimeter is first carried on SARAL/AltiKA (SRL) (CNES, 2016a). Ka-band altimeter data are different with Ku-band data, which has been proved by several researches, including absolute calibrations, observation assessments, retracking methods, geoid derivation and gravity anomaly recovery (Babu et al., 2015; Smith, 2015; Zhang and Sandwell, 2017; Zhu et al., 2020; Zhu et al., 2021). However, Ka-band data are hardly specifically processed in the construction of the recognized global marine gravity models. Moreover, HY-2A, China's first ocean dynamical satellite, was launched on August 16, 2011. A microwave imager, a dual-frequency (Ku band and C band) radar altimeter and a Ku-band scatterometer on HY-2A are used to obtain brightness temperature, monitor basic ocean elements (sea level, significant wave height and wind speed) and determine sea surface vector wind field. Radar altimeter on HY-2A has perform geodetic mission for about four years. HY-2A has been proved to play an important role in determining deflections of vertical (DOVs) and recovering gravity anomalies (Rapp, 1979; Zhu et al., 2019; Wan et al., 2020; Ji et al., 2021a; Guo et al., 2022). However, HY-2A-measured altimeter data are rarely used for published global models of gravity anomalies.

Accuracy of altimeter-derived gravity anomalies in offshore waters is low because of the waveform contamination by land. Compared with traditional Ku/C-band altimeters, Ka-band altimeter with higher frequency has a smaller altimeter footprint (CNES, 2016a), which leads to the smaller contamination radius of land. Moreover, the gravity anomaly model derived from more altimeter data is more accurate. For the recognized marine gravity anomaly models, HY-2A-measured altimeter data are not used, and Ka-band data are hardly specifically processed. Therefore, we will construct the global marine gravity anomaly model (SDUST2021GRA) on a $1' \times 1'$ grid from multi-satellite altimeter data including HY-2A-measured data. In the processing, noise variance of Ka-band along-track geoid gradients is determined by the different method from those of Ku-band observations.

First, along-track geoid gradients are calculated from altimeter-measured sea surface heights (SSHs). Second, gridded DOVs are determined by the least-squares collocation (LSC) method (Rapp, 1979). Final, gravity anomalies are derived from gridded DOVs by the inverse Vening-Meinesz formula (IVM) (Hwang, 1998). In the process of calculating gridded DOVs, the noise variance of Ka-band along-track geoid gradients for GM in LSC is determined by the iteration method which is proposed by Zhu et al. (2020). In Section 2, the research area and data are introduced. In Section 3, the methods of data preprocessing, calculating gridded DOV and derived gravity anomalies are presented in detail, respectively. In Section 4,

the global marine gravity model is analyzed by comparing with other models. Meanwhile, the accuracy of the model is assessed by shipborne gravity data. The conclusion is given in Section 5.

## 2 Research data and area

### 2.1 Study area

The ocean covering 0°~360°E and 80°S~80°N is selected as the study area. The study area is divided into 144 regions to derive gravity anomalies due to the limited memory of the computer (Figure 1). From 0° to 360°E, regions are marked from L1 to L18; from 80°S to 80°N, regions are marked from B1 to B8.

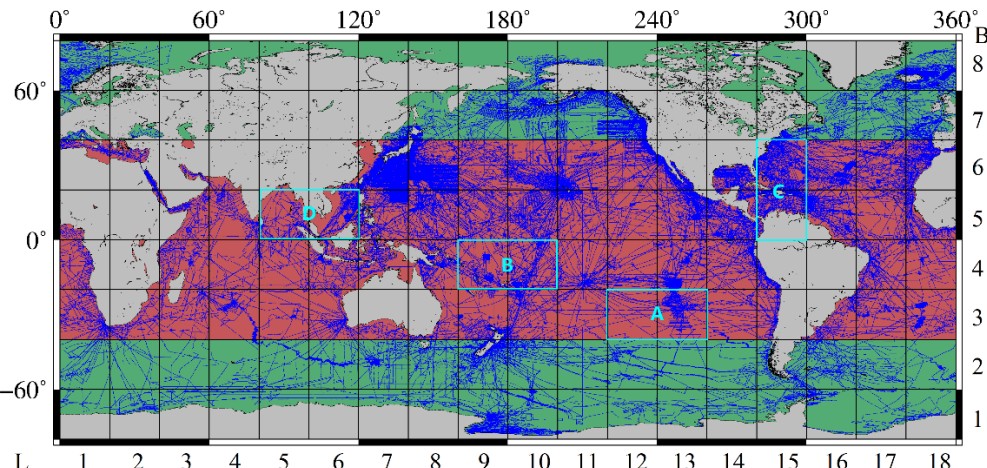

Figure 1. Division map for deriving gravity anomalies and tracks of NCEI shipborne data. The red and green areas means that the reference gravity anomalies are obtained from XGM2019e and EGM2008, respectively. The lines in blue present the cruises of shipborne gravity. The areas in cyan boxes are the special areas for analysis. From 0° to 360°E, regions are marked from L1 to L18; from 80°S to 80°N, regions are marked from B1 to B8.

### 2.2 Altimeter Data.

Non-time critical Level 2 Plus (L2P) Version 3.0 products of altimeter data released by archiving validation and interpretation of satellite oceanographic data (AVISO) (ftp://ftp-access.aviso.altimetry.fr) are used to construct the gravity anomaly model. The reference ellipsoid used for L2P Version 3.0 products is the World Geodetic System (WGS) 84 reference ellipsoid. The L2P products at the 1-Hz sampling frequency are along-track products that contain only data relating to marine surfaces. They have a homogenized format and content for all altimeter missions (CNES, 2020). Since L2P products of Jason-2 for GM have only a few cycles of data, the altimeter data at the 1-Hz sampling frequency from geophysical data records (GDRs) of Jason-2/GM are used.

The accuracy of SSHs from altimetry data is gradually improved through the years. ERS-1 was launched before 1990, and ERS-1/GM-measured altimeter data play little role in deriving gravity anomalies from multi-satellite altimeter data (Zhu et al., 2020; Sandwell et al., 2021). Therefore, the GM data used for constructing the gravity model only contain the altimeter data of satellites launched after 1990, as listed in Table 1. Although the exact repeat mission (ERM) provides the sparse track coverage, accurate average SSHs can be obtained from long-term ERM data. The ERM data in Table 1 are also used to derive gravity anomalies.

For Topex/Poseidon (T/P), Jason-1, Jason-2 and Jason-3, satellites are firstly located on their nominal orbit for an ERM, and then swift to the interleaved orbit for the other ERM (CNES, 2016b, 2017, 2021). The first ERM is marked with an '_A' after the satellite name, e.g., T/P_A, and the second ERM is marked with an '_B' after the satellite name (Table 1). Envisat is marked in the same way.

In order to ensure the continuity of gravity anomalies between regions, altimeter data in the areas extending outward 1 degree from these regions are used for deriving gravity, e.g., altimeter data in the area of 59°~81°E and 41°S~19°S are used for deriving gravity in the region of L4B3 (60°~80°E and 40°S~20°S).

Table 1. Information of altimetry satellites used for deriving gravity field

| Mission | Satellite | Period | Cycles | Latitude range (°) | Cycle duration (d) | Inter-track distance at equator (km) |
|---|---|---|---|---|---|---|
| Geodetic mission | Jason-1 | 12.05-13.06 | 500-537 | ±66 | 406 | ~7.5 |
| | HY-2A | 16.03-20.06 | 118-288 | ±81 | 168 | ~15 |
| | CryoSat-2 | 10.07-20.05 | 007-130 | ±88 | 369 | ~2.5 |
| | SRL | 16.07-20.08 | 100-142 | ±81.5 | - | ~5 |
| | Jason-2 | 17.07-19.10 | 500-537/ 600-644 | ±66 | 369 | ~7 |
| Exact repeat mission | T/P_A | 92.09-02.08 | 001-364 | ±66 | 10 | ~315 |
| | Jason-1_A | 02.01-09.01 | 001-259 | | | |
| | Jason-2_A | 08.07-16.10 | 001-303 | | | |
| | Jason-3_A | 16.02-20.09 | 001-169 | | | |
| | T/P_B | 02.09-05.09 | 369-479 | | | |
| | Jason-1_B | 09.02-12.03 | 262-374 | | | |
| | Jason-2_B | 16.10-17.05 | 305-327 | | | |
| | Envisat_A | 02.05-10.10 | 006-093 | ±81.5 | 35 | ~80 |
| | Envisat_B | 10.11-12.04 | 097-113 | | | |
| | HY-2A | 14.04-16.03 | 067-117 | ±81.5 | 14 | ~210 |
| | SRL | 13.03-15.03 | 001-021 | ±81.5 | 35 | ~80 |

**2.3 Gravity data and other data**

**2.3.1 Reference gravity anomalies**

Gravity anomalies on regular grids can be obtained by the calculation function of gravity field functionals on ellipsoidal grids provided by International Centre for Global Earth Models (ICGEM) (http://icgem.gfz-potsdam.de/calcgrid). Earth

Gravitational Field Model 2008 (EGM2008) (Pavlis et al., 2012) and XGM2019e (Zingerle et al., 2020) are the recognized high-precision Earth Gravitational Field models.

EGM2008 is complete to degree and order 2159, with additional coefficients up to degree 2190 and order 2159. Its half-wavelength resolution is about 9 km. XGM2019e (Zingerle et al., 2020) is a combined global gravity field model complete to degree and order 5399. Its half-wavelength resolution is about 4 km. XGM2019e is mainly constructed from the GOCO06s satellite-only gravity field model, 15′ ground gravity dataset provided by the US National Geospatial-Intelligence

Agency and 1′ augmentation dataset.

Since ICGEM only provides the calculation for XGM2019e up to degree and order 2159 (marked as the XGM2019e_2159 model), XGM2019e up to degree and order 2159 is used as the reference gravity field model. Compared with shipborne gravity anomalies, the differences for XGM2019e are greater than those for EGM2008 in some sea areas of the middle to

120 high latitudes. Therefore, EGM2008 up to the order 2160 is used as the reference gravity field model in the areas between 40°S~80°S or 40°N~80°N referring to previous studies (Sandwell et al., 2014a; Shih et al., 2015). As shown in Figure 1, the reference gravity field model of the regions in red is XGM2019e up to degree and order 2159, and that in green is EGM2008 up to the order 2160.

**2.3.2 Altimeter-derived gravity anomaly model**

According to the altimetry data used for deriving SDUST2021GRA, we select the model SIO V30.1 (Sandwell et al., 2021) on 1′×1′ grids established from altimeter data in the similar period released by the Scripps Institution of Oceanography (SIO). The marine gravity model of DTU17 (Andersen and Knudsen, 2019) released by Technical University of Denmark (DTU) is also used to be compared with SDUST2021GRA. The GM altimeter data in Table 2 are used in the global marine gravity

models.

**Table 2. Altimeter GM data used for deriving marine gravity anomaly models (Unit: month)**

|  | Geosat | ERS-1 | Jason-1 | Jason-2 | CryoSat-2 | SRL | HY-2A |
|---|---|---|---|---|---|---|---|
| DTU17 | 18 | 12 | 14 | 0 | ~84 | ~12 | 0 |
| SIO V30.1 | 18 | 12 | 14 | 20 | 114+ | 50+ | 0 |
| SDUST2021GRA | 0 | 0 | 14 | 20 | 118 | 49 | 51 |

**2.3.3 Shipborne gravity anomalies**

The shipborne gravity data for assessing the accuracy of gravity models are shown in blue in Figure 1, which are provide by US National Centers for Environmental Information (NCEI) (https://www.ncei.noaa.gov/maps/geophysics/). There are about

135 2000 cruises which are measured by different instruments from different countries and institutions in different time; a feature that requires some processing.

### 2.3.4 Mean dynamic topography model

Geoid heights can be calculated from SSHs by subtracting dynamic topography. However, accurate dynamic topography is difficult to obtain, so the mean dynamic topography (MDT) model is used in the paper. MDT-CNES-CLS18 (Mulet et al., 2021) is the most updated MDT model released by AVISO (https://www.aviso.altimetry.fr/en/data/products/auxiliary-products/mdt/mdt-global-cnes-cls18.html), which is a global model on 0.125°×0.125° grids of differences between mean sea level heights and geoid heights from 1993 to 2012. The data used for establishing the model mainly include the mean sea surface model of CNES-CLS15 (Pujol et al., 2018), the geoid model of GOCO05S (Mayergürr et al., 2015), hydrological data and drifter data.

### 3 Methodology

### 3.1 Preprocessing Method

### 3.1.1 Shipborne data preprocessing

Since shipborne gravity data in different reference systems provided by NCEI were measured by different organizations, the reference datum of shipborne data should be unified. Moreover, there are some long-wavelength errors in shipborne gravity, which are caused by drifts in gravimeter readings, off-leveling, incorrect ties to base stations, and different reference fields (Wessel and Watts, 1988).

First, the gross errors are excluded by the 3 sigma rules. The mean value and STD of gravity anomalies for each cruise are calculated. Mean removal gravity anomalies are obtained from gravity anomalies by subtracting the mean value. If the difference between the mean removal gravity anomaly and reference gravity anomaly at a point is greater than three times of the STD, the observation of the point is rejected.

Then, the quadratic polynomial is used for unifying the gravity reference datum and correcting long-wavelength errors (Hwang and Parsons, 1995; Guo et al., 2022). The differences between gravity anomalies from the reference gravity field model and those from NCEI can be presented by

$$\Delta dg_i(t) = a_i + b_i \Delta t + c_i \Delta t^2 \tag{1}$$

where $i$ is the ID of a shipborne cruise, and $\Delta dg_i$ are the differences between reference and shipborne gravity anomalies at the observation points of the cruise. $\Delta t$ is computed from the observation time $t$ minus the departure time. $a_i$, $b_i$ and $c_i$ are parameters obtained, for each cruise, by least-square fitting from Equation (1).

Finally, corrected shipborne gravity anomalies can be obtained from original shipborne gravity anomalies by adding corrections. The shipborne gravity anomaly discrepancies at crossovers of different cruises are obviously decreased after the adjustment than those before the adjustment (Zhu et al., 2019; Ji et al., 2021b; Guo et al., 2022).

### 3.1.2 Altimeter data preprocessing

There are some errors in SSH observations of altimeter data, including instrument errors, propagation errors and geophysical errors. Corrections for the errors are provided in L2P products. The final SSHs are calculated from original SSHs plus the corrections. The reference ellipsoid used for Jason-2/GM altimeter data is different from WGS84 used for L2P products. The reference ellipsoid is the first-order definition of the non-spherical shape of Earth with equatorial radius of 6378.1363 km and flattening coefficient of 1/298.257 (CNES, 2017), named T/P ellipsoid. Jason-2/GM-measured SSHs in the T/P ellipsoid should be transformed to those in WGS84 ellipsoid by

$$
\begin{aligned}
B_w &= B + \frac{N}{(M+h)^2} e^2 \sin B \cos B \mathrm{d}a + \frac{N(2 - e^2 \sin^2 B)}{(M+h)(1-\alpha)} \sin B \cos B \mathrm{d}\alpha \\
L_w &= L \\
h_w &= h - \frac{N}{a}(1 - e^2 \sin^2 B)\mathrm{d}a + \frac{M}{(1-\alpha)}(1 - e^2 \sin^2 B)\sin^2 B \mathrm{d}\alpha
\end{aligned}
\tag{2}
$$

where, $\mathrm{d}a$ is the difference between semimajor axis of WGS84 ellipsoid and that of T/P ellipsoid, and $\mathrm{d}\alpha$ is the difference between flattening of the two ellipsoids. $a$ and $e$ are the semimajor axis and first eccentricity of T/P ellipsoid. $B$, $L$ and $h$ are the latitude, longitude and SSH in the T/P ellipsoid, respectively. $B_w$, $L_w$ and $h_w$ are the corresponding data in the WGS84 ellipsoid. $N$ and $M$ are the radius of curvature in prime vertical and meridian in the T/P ellipsoid.

Sea surface temporal variability and high-frequency noise affect the accuracy of SSHs, so Gaussian filtering is used for the along-track GM-measured SSHs. The corresponding response function is

$$
f(r) = \exp(-\frac{S^2}{2r_c^2})
\tag{3}
$$

where, $S$ is the sphere distance between substellar points, and $r_c$ is the radius of the convolution window. Following our previous study (Zhu et al., 2020), the value of $r_c$ is 7 km in the paper.

The amount of ERM-measured SSHs is too large to be processed in the same way with GM-measured SSHs. Due to the repeated tracks, the ground track control band of ERM is 1~2 km. The simplified collinear adjustment method (Figure 2) is used to reduce high-frequency noise of ERM-measured SSHs (Jin et al., 2016; Yuan et al., 2020). First, the track with the largest amount of observed data among all repeated tracks is selected as the reference track. Second, SSHs of other tracks are interpolated into the reference track. Final, the interpolated SSHs are averaged to obtain the mean SSH on the reference track.

In Figure 2, $R_i$ is the observation point on the reference track, and $P_k$ and $P_{k+1}$ are the observation points on the other track. The SSH on point $R_i'$ whose latitude is the same with $R_i$ can be interpolated from SSHs on points $P_k$ and $P_{k+1}$. The mean value of SSHs on $R_i$, $R_i'$ and interpolated points on other tracks is the adjusted SSH on $R_i$ of the reference track.

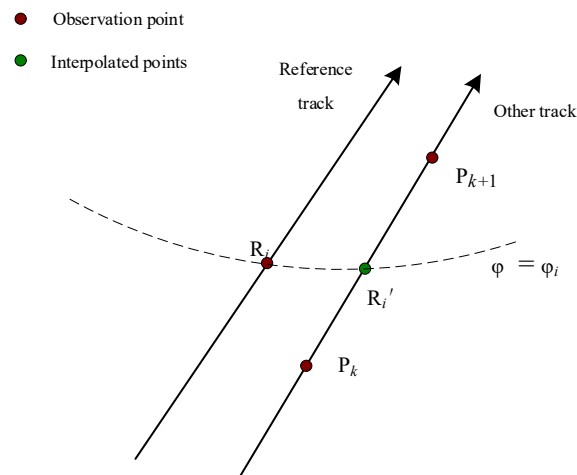

**Figure 2. Simplified collinear adjustment method. $R_i$ is the observation point on the reference track. $P_k$ and $P_{k+1}$ are the observation points on the other track. The latitude of $R_i'$ is the same as that of $R_i$. The dashed line indicates the parallel.**

## 3.2 Method of gridding DOV

### 3.2.1 Along-track geoid gradient calculation

Dynamic topography is the difference between the geoid height and the SSH. Therefore, the geoid height is calculated from the adjusted SSH minus the MDT (from MDT-CNES-CLS18). Based on the remove-restore method, geoid heights from SSHs minus those from the reference gravity model are residual geoid heights (Sansò and Sideris, 2013). Then, the residual along-track geoid gradient is

$$e_{res} = \frac{dN_{res}}{dS} \tag{4}$$

where, $dS$ is the sphere distance between two adjacent substellar points, and $dN_{res}$ is the difference between residual geoid heights at the two points.

### 3.2.2 Gridded DOV calculation

The LSC method has been proved useful in DOV determination, so DOV on regular grids can be obtained by

$$\begin{pmatrix} \xi_{res} \\ \eta_{res} \end{pmatrix} = \begin{pmatrix} C_{\xi e} \\ C_{\eta e} \end{pmatrix} (C_{ee} + C_n)^{-1} e_{res} \tag{5}$$

where, $\xi_{res}$ and $\eta_{res}$ are the residual meridian and prime vertical components of the DOV, respectively. $C_{ee}$ is the covariance matrix for $e_{res}$. $C_{\xi e}$ and $C_{\eta e}$ are the covariance matrices for $\xi_{res}$-$e_{res}$ and $\eta_{res}$-$e_{res}$, respectively. $C_n$ is the diagonal matrix of noise variances of along-track geoid gradients.

As covariance functions of disturbing potentials are isotropic, the covariance function of residual disturbing potentials at the given distance can be calculated by (Tscherning and Rapp, 1974; Hwang, 1989)

$$K_{res}(P,Q) = \sum_{n=2}^{N_{max}} \delta\sigma_n s^{n+1} P_n(\cos\psi_{PQ}) + \sum_{n=N_{max}+1}^{\infty} \sigma_n s^{n+1} P_n(\cos\psi_{PQ}) \tag{6}$$

where, $N_{max}$ is the maximum degree of the reference gravity model, and $s$ is obtained from mean radius of Bjerhammar sphere and Earth sphere. $\sigma_n$ is the degree variance of disturbing potentials, which is calculated based on Model 4 proposed by Tscherning and Rapp (1974). $\delta\sigma_n$ is the error degree variance of disturbing potentials (Hwang, 1989), which is obtained from errors of coefficients in the potential set of the reference gravity model.

As all data related to gravity can be expressed as functionals of disturbing potentials, covariance functions of residual DOV components can be calculated from the covariance function of residual disturbing potentials. The covariance functions of deflection components $\xi$ and $\eta$ are not isotropic, but the longitude and transvers components are isotropic. The geoid gradient has the same value and opposite sign as the DOV, so covariance functions of longitude components $l$ and transvers components $m$ of the residual geoid ($C_{ll}$, $C_{mm}$) have simple relations to $K_{res}(P,Q)$. Therefore, $C_{ee}$, $C_{\xi e}$ and $C_{\eta e}$ can be obtained from $C_{ll}$ and $C_{mm}$ by

$$\begin{aligned} C_{ee} &= C_{ll}\cos(\alpha_{e_P}-\alpha_{PQ})\cos(\alpha_{e_Q}-\alpha_{PQ}) + C_{mm}\sin(\alpha_{e_P}-\alpha_{PQ})\sin(\alpha_{e_Q}-\alpha_{PQ}) \\ C_{\xi e} &= -C_{ll}\cos\alpha_{PQ}\cos(\alpha_{e_Q}-\alpha_{QP}) + C_{mm}\sin\alpha_{PQ}\sin(\alpha_{e_Q}-\alpha_{QP}) \\ C_{\eta e} &= -C_{ll}\sin\alpha_{PQ}\cos(\alpha_{e_Q}-\alpha_{QP}) - C_{mm}\cos\alpha_{PQ}\sin(\alpha_{e_Q}-\alpha_{QP}) \end{aligned} \tag{7}$$

where, $\alpha_{e_P}$ and $\alpha_{e_Q}$ are azimuths of ground track at points $P$ and $Q$, respectively. $\alpha_{PQ}$ is the azimuth from $P$ to $Q$, and $\alpha_{QP}$ is that from $Q$ to $P$.

### 3.2.3 Noise variances of Ka-band and Ku-band geoid gradients

As $C_{ee}$, $C_{\xi e}$, $C_{\eta e}$ and $e_{res}$ in Equation (5) can be obtained referring to Section 3.2.2, noise variances of along-track geoid gradients are needed for calculating gridded DOVs. The noise variance of SSHs can be obtained by calculating the STD of

SSHs at 20-Hz (40-Hz for SRL) sampling frequency, but there are no 20-Hz SSH data in L2P products. Therefore, SSH crossover discrepancies are used to assess the accuracy of SSHs.

Since residual along-track geoid gradients are obtained by Equation (4), the difference between SSHs at two adjacent points can effectively weaken the effect of long wavelength errors of SSHs on geoid gradients (Mcadoo et al., 2008), e.g., satellite orbit errors, propagation errors, and dynamic topography errors. SSH crossover adjustment can reduce the radial orbit errors (Yuan et al., 2021). Ignoring errors of distance between two adjacent ground points, noise variances of along-track geoid gradients are computed from crossover discrepancies of SSHs after the crossover adjustment by

$$D_e = \frac{2D_{SSH}}{dS^2} = \frac{D_{\Delta SSH}}{dS^2} \tag{8}$$

where $D_{SSH}$ is the covariance of adjusted SSHs, and $D_{\Delta SSH}$ is the covariance of crossover discrepancies of adjusted SSHs.

In offshore waters covering 0°~30°N and 105°E~125°E, the accuracy of SSHs from SRL is improved by about 10% compared with that from HY-2A, and the accuracy of along-track geoid gradients is improved by about 30%. This is because that SSHs from Ka-band altimeter are more sensitive to rainy and cloudy conditions than those from Ku-band to have larger propagation errors. The differentiation in Equation (4) can effectively weaken the effects of propagation errors on along-track geoid gradients. However, the SSH crossover adjustment cannot effectively reduce propagation errors, so the iteration method for assessing accuracy of along-track geoid gradients from SRL is proposed by Zhu et al. (2020).

The precision of along-track geoid gradients in Ku-band GM missions (Jason-1/GM, Jason-2/GM, CryoSat-2 and HY-2A/GM) is assessed by the Equation (8) from crossover discrepancies of SSHs after the crossover adjustment. Gridded DOVs of each satellite are determined by Equation (5), which are used to derive gravity anomalies by IVM method presented in Section 3.3. The accuracy of altimeter-derived gravity anomalies can be assessed by shipborne gravity data and SIO V30.1 (Zhu et al., 2020).

The relationship among the precision of altimetric gravity, precision of geoid gradients, and density of geoid gradients can be presented by (Zhu et al., 2020)

$$D_{\Delta\hat{g}} = \beta_0 + \beta_1 \frac{\rho}{D_e} \tag{9}$$

where, $D_{\Delta\hat{g}}$ is the variance of altimetric gravity, and $\rho$ is the average number of along-track geoid gradients in 1′×1′ area. Parameters of $\beta_0$ and $\beta_1$ can be calculated by the LS fitting method.

The accuracy of along-track geoid gradients of SRL/DP can be obtained following the method in Figure 3. First, initial precision of along-track geoid gradients of SRL/DP is also assessed by the Equation (8). Gravity anomalies are derived from SRL/DP-measured SSHs based on the initial precision. Second, the precision of SRL/DP-derived gravity is assessed by shipborne gravity and SIO V30.1 model, then used to calculate the new precision of along-track geoid gradient from
SRL/DP by Equation (9). The new SRL/DP gravity anomalies are derived based on the new precision of along-track geoid gradients. Final, the calculation of step second is repeated, and terminated when the difference in precision of altimetric gravity between adjacent times is less than 0.02 mGal.

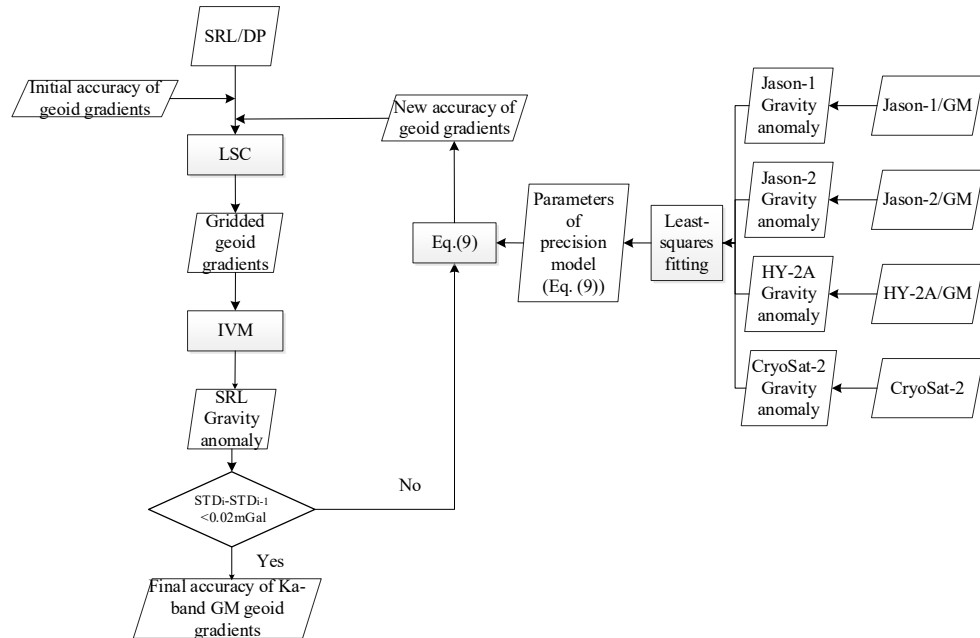

**Figure 3. Iterative method of assessing the accuracy of SRL/DP-measured along-track geoid gradients. Initial accuracy of along-track geoid gradients is only used for the first calculation of gravity, and new accuracy is used for other calculations.**

As the simplified collinear adjustment is used for altimeter data in ERM, the influence of cloud and rain condition on adjusted SSHs of SRL/ERM is weakened by the average calculation. Therefore, the accuracy of Ka-band along-track geoid gradients in ERM can be directly determined by crossover discrepancies.

In conclusion, the precision of along-track geoid gradients in Ku band for all missions and in Ka band for ERM is assessed by Equation (8) from crossover discrepancies of SSHs after the crossover adjustment, and that in Ka band for GM is assessed by the iteration method.

## 3.3 Method of deriving gravity anomalies

Vening-Meinesz Formula can be used to determine DOVs from gravity anomalies, so the inverse of Vening-Meinesz formula is used to derive gravity anomalies from DOVs with the development of altimetry technology (Hwang, 1998; Ji et al., 2021a). Hwang (1998) derived the inverse Vening-Meinesz formula and the kernel function based on the spherical harmonic expansion of disturbing potential (Heiskanen and Moritz, 1967), Green's formula (Meissl, 1971), Laplace surface operator (Courant and Hilbert, 1953), orthogonality relationship of fully normalized spherical harmonics (Heiskanen and

Moritz, 1967) and the kernel function by Meissl (1971). Gravity anomalies can be derived by inverse Vening-Meinesz formula:

$$\Delta g(p) = \frac{\gamma_0}{4\pi} \iint_{\sigma} H'(\psi)\left(\xi_q \cos\alpha_{qp} + \eta_q \sin\alpha_{qp}\right) d\sigma_q \tag{10}$$

where, $\gamma_0$ is the normal gravity at point p. $H'(\psi)$ is the kernel function relating to the sphere distance $\psi$ between points $p$ and $q$, which is obtained by

$$H' = -\frac{\cos\dfrac{\psi}{2}}{2\sin^2\dfrac{\psi}{2}} + \frac{\cos\dfrac{\psi}{2}\left(2\sin\dfrac{\psi}{2}+3\right)}{2\sin\dfrac{\psi}{2}\left(\sin\dfrac{\psi}{2}+1\right)} \tag{11}$$

As the meridian and prime vertical components of DOVs are given on a regular grid, one-dimensional fast Fourier transform (1D-FFT) is used for the calculation of IVM formula, that is, gravity anomalies at the same parallel are computed simultaneously.

The kernel function $H'(\psi)$ is singular when the distance $\psi$ is zero, the innermost zone effect should be considered. The shape of the innermost zone is assumed as a circle, a square and a rectangle in different researches (Hwang, 1998; Li et al., 2018). In this study, we use the circular innermost zone following Hwang (1998).

## 4 Gravity anomaly results

Remove-restore method is used for constructing the global marine gravity anomaly model. Following the method (Figure 4)

presented in Section 3, gridded residual DOVs are determined from along-track geoid gradients by LSC method, in which the precision of along-track geoid gradients in Ka band for GM is assessed by the iteration method (Figure 3) and that of the other along-track geoid gradients is assessed by crossover discrepancies of SSHs. Then, residual gravity anomalies are derived from gridded residual DOVs by 1D-FFT based on IVM formula. The global marine gravity anomaly model (SDUST2021GRA) in Figure 5 is established from the residual gravity anomalies by restoring the reference gravity

anomalies.

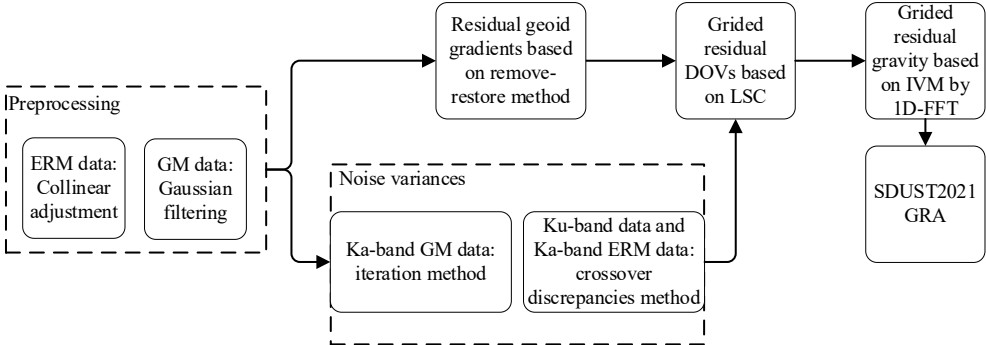

**Figure 4. Method for establishing ocean gravity anomaly model from multi-satellite altimeter data**

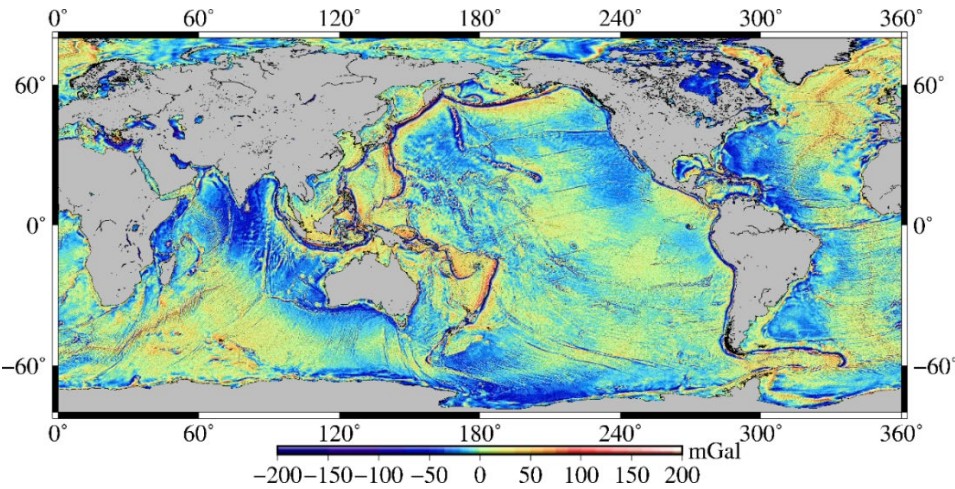

**Figure 5. The marine gravity anomaly model of SDUST2021GRA**

## 4.1 Comparison with SIO V30.1 and DTU17

Recognized marine gravity anomaly models of SIO V30.1 and DTU17 are used for verify the reliability of SDUST2021GRA. The differences between SDUST2021GRA and recognized models are shown in Figure 6. The differences in coastal areas (e.g., the Aleutian Islands and the Philippine Islands) are greater than those in open oceans, which is caused by the waveform contamination by land and islands. Moreover, the differences in areas with rapidly changing submarine topography are greater than those in areas with flat submarine topography, e.g., the South Sandwich Trench in Figure 7. Therefore, gravity anomaly models can be used for construction of submarine topography models. Moreover, compared with SDUST2021GRA, the differences for DTU17 in Figure 6(b) are smaller than those for SIO V30.1 in Figure 6(a). The distribution of differences between DTU17 and SIO V30.1 in Figure 6(c) is similar to that between SDUST2021GRA and SIO V30.1.

Histograms of the differences between altimeter-derived gravity anomaly models are shown in Figure 8. In Figure 8(a), the differences between DTU17 and SIO V30.1 are mainly concentrated between -5 mGal and 5 mGal, which accounts for about 95% of the total number. The distribution of differences between SDUST2021GRA and SIO V30.1 in Figure 8(b) is similar to that in Figure 8(a). In Figure 8(c), compared with DTU17, the differences for SDUST2021GRA between -5 mGal and 5 mGal account for about 98% of the total number, and those between -3 mGal and 3 mGal account for about 93%. This also shows that compared with SDUST2021GRA, the differences for DTU17 are smaller than those for SIO V30.1. Based on the small differences between SDUST2021GRA and recognized models, we can conclude that SDUST2021GRA is reliable.

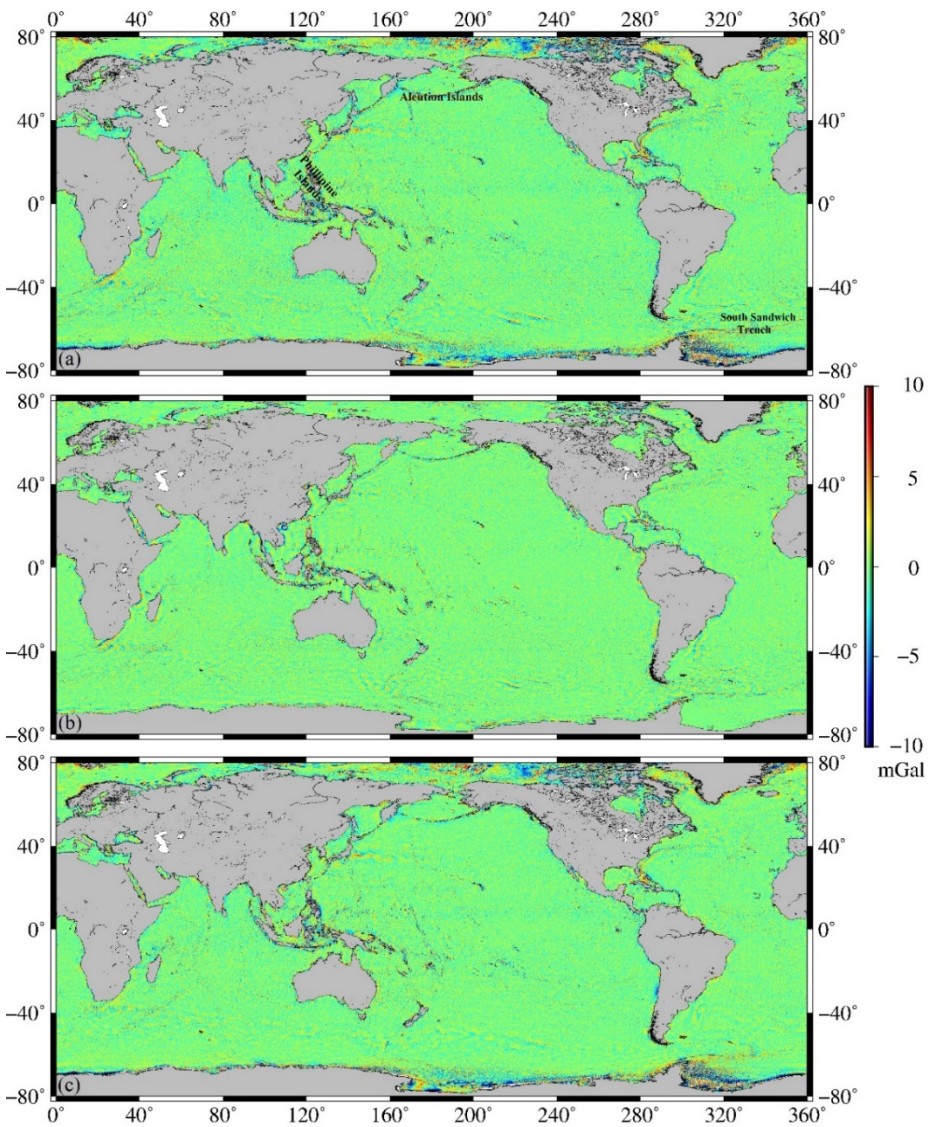

**Figure 6. Different between SDUST2021GRA and recognized marine gravity models: (a) is for SDUST2021GRA and SIO V30.1, (b) is for SDUST2021GRA and DTU17, and (c) is for DTU17 and SIO V30.1.**

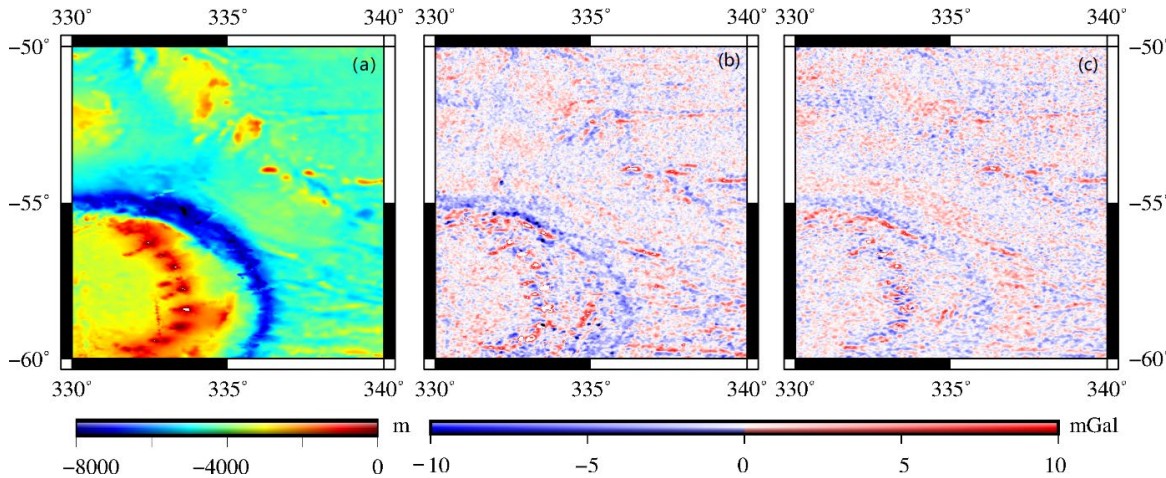

**Figure 7. Topography and gravity information around South Sandwich Trench: (a) is the submarine topography, (b) is the figure of differences between SDUST2021GRA and SIO V30.1, and (c) is for SDUST2021GRA and DTU17.**

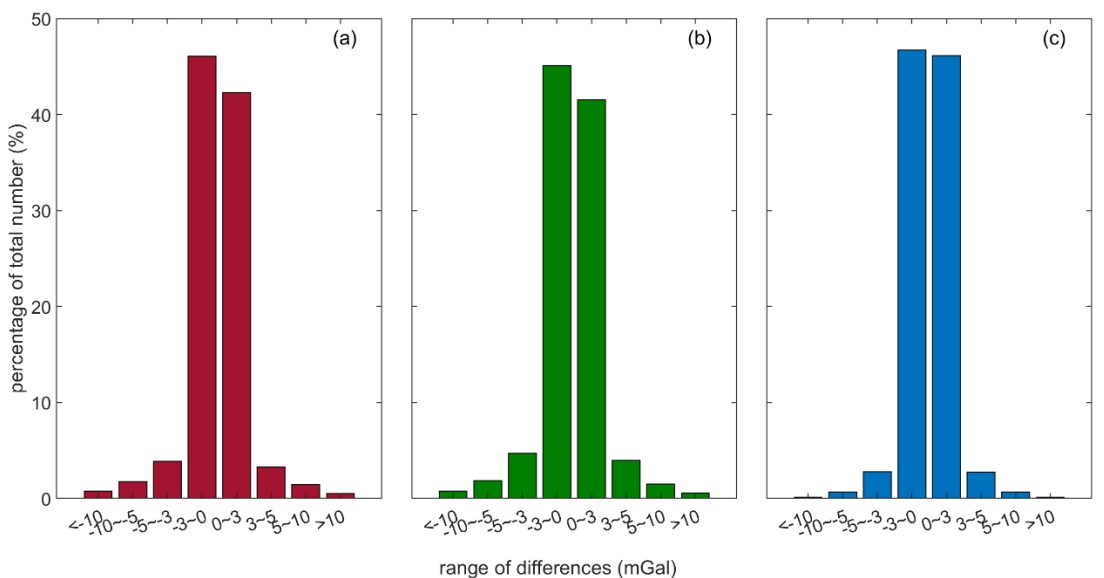

**Figure 8. Histograms of different between altimeter-derived oceanic gravity anomaly models: (a) is for DTU17 and SIO V30.1, (b) is for SDUST2021GRA and SIO V30.1, and (c) is for SDUST2021GRA and DTU17.**

## 4.2 Shipborne gravity data assessment

Shipborne gravity data are adjusted by the quadratic polynomial based on the reference gravity model. It can be considered that the adjusted shipborne data are independent of altimeter-derived gravity models, so the shipborne data are used to assess the accuracy of gravity models.

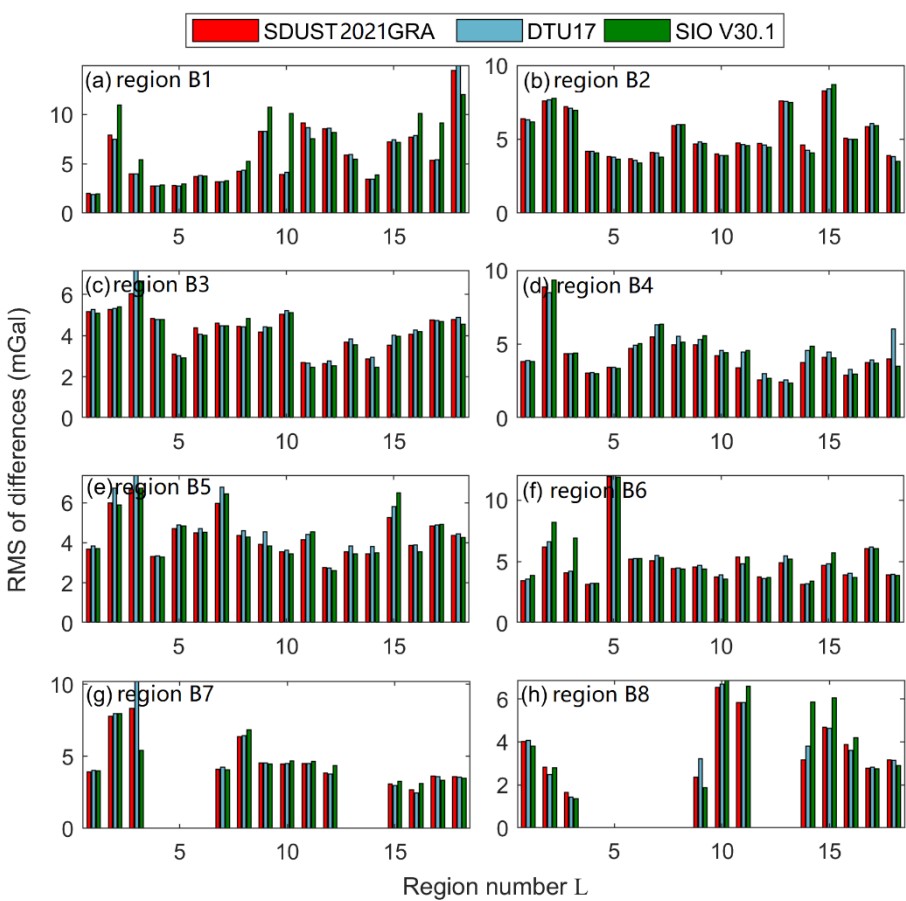

**Figure 9. RMS of differences between altimeter-derived gravity models and shipborne data: Figures from (a) to (h) present the RMSs in areas from B1 to B8.**

First, the RMSs of differences between altimeter-derived gravity models and shipborne data in the 144 regions in Figure 1
are listed in Table A1 and shown in Figure 9. In Figure 9, there is no data in region L4B7, L5B7, L6B7 and L13B7, which is
because that these areas have no sea.  There is no data in other regions in Figure 9, which is caused by no shipborne gravity
data in these areas.

We can see that the three models show different levels of accuracy in different regions. In order to further compare the
accuracy of each model, The differences between RMSs for the three models in Figure 9(b) in each region are calculated
(Figure 10(a)). If the difference is greater than 0, it means that the accuracy of the former model is lower than that of the
latter model. Therefore, In the 18 regions marked B2, the accuracy of SIO V30.1 are higher than that of SDUST2021GRA in
13 regions, and higher than that of DTU17 in 15 regions. Moreover, the differences between RMSs for the three models in 8
regions marked L15 (region L15 in Figure 9(a)~9(h)) are shown in Figure 10(b). The accuracy of SDUST2021GRA is higher
than that of SIO V30.1 in 6 regions, and higher than that for DTU17 in 6 regions. It can be seen from Figure 1 that the

regions marked B2 are mainly the open sea areas and the regions marked L15 have complex coastlines. Therefore, the accuracy of SDUST2021GRA is slightly higher than that of DTU17 and SIO V30.1 in sea areas with complex coastlines and islands. In the open ocean, SIO V30.1 has the best accuracy.

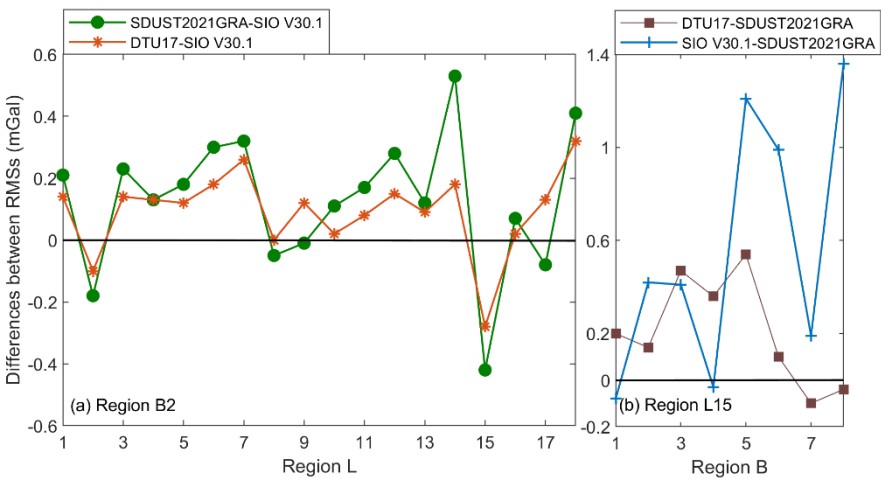

Figure 10. Differences between RMSs. The RMSs are statistics of differences between altimeter-derived gravity models and shipborne data: (a) for region L1B2, L2B2,···, L18B2; (b) for region L15B1, L15B2,···, L15B8.

Second, four typical ocean areas, marked as A ~D, are selected for analyzing the accuracy of altimeter-derived models, as shown in Figure 1. Area A is the open ocean without special submarine topography. There are many islands in area B, including Solomon Islands, Tuvalu Islands, Phoenix Islands and Cook Islands. Area C and D have the complex coastline and many islands. Compared with shipborne gravity data, the statistical information of differences for altimeter-derived gravity models is listed in Table 3. Assessed by the gravity crossover discrepancies after excluding the value greater than 20 mGal, accuracy of shipborne data in different areas is listed in Table 4. The accuracy of altimeter-derived gravity models is calculated by the law of error propagation(Table 4).

As listed in Table 3 and Table 4, the accuracy of SIO V30.1 is highest in area A, and that of SDUST2021GRA is highest in area B, area C and area D. These indicate that SIO V30.1 has the best accuracy in the open ocean and SDUST2021GRA has the best accuracy in the offshore areas and the areas with many islands. The conclusion is consistent with the analysis of altimeter-derived models by Figure 9 and Figure 10. Moreover, accuracy of SDUST2021GRA in the open oceans and areas with many islands are better than 2 mGal, which is consistent with that of modern shipborne gravity (1~2 mGal) (Ling et al., 2021).

HY-2A-measured altimeter data are excluded from the multi-satellite altimeter dataset, and the residual altimeter data are used to derived gravity anomalies marked SDUST(no HY-2A), as listed in Table 4. Compared with the accuracy of

SDUST2021GRA, that of SDUST(no HY-2A) reduces by 3.8%, 1.2% and 2.7% in Area B, C and D, respectively. These indicate that HY-2A has an important role in gravity anomaly recovery in areas with complex coastline and many islands.

**Table 3. Statistics of differences between altimeter-derived and shipborne gravity data in different areas (Unit: mGal)**

|  |  | Min | Max | Mean | STD | RMS |
|---|---|---|---|---|---|---|
|  | SDUST | -37.15 | 41.87 | -0.01 | 3.58 | 3.58 |
| Area A | DTU | -36.62 | 41.08 | 0.00 | 3.72 | 3.72 |
|  | SIO | -37.55 | 41.70 | 0.13 | 3.44 | 3.44 |
|  | SDUST | -43.20 | 49.41 | -0.05 | 4.43 | 4.43 |
| Area B | DTU | -46.33 | 50.66 | -0.17 | 4.77 | 4.77 |
|  | SIO | -71.40 | 52.86 | -0.08 | 4.76 | 4.76 |
|  | SDUST | -52.05 | 48.19 | 0.01 | 4.97 | 4.97 |
| Area C | DTU | -51.15 | 50.14 | 0.07 | 5.30 | 5.30 |
|  | SIO | -70.85 | 114.21 | 0.43 | 6.06 | 6.07 |
|  | SDUST | -48.73 | 40.96 | 0.01 | 4.53 | 4.53 |
| Area D | DTU | -49.24 | 42.96 | -0.04 | 4.73 | 4.73 |
|  | SIO | -49.65 | 46.83 | 0.32 | 4.57 | 4.58 |

**Table 4. Accuracy of altimeter-derived and shipborne gravity data in different areas (Unit: mGal)**

|  | Area A | Area B | Area C | Area D |
|---|---|---|---|---|
| SDUST | 1.39 | 1.82 | 3.34 | 2.22 |
| DTU | 1.72 | 2.54 | 3.81 | 2.60 |
| SIO | 0.97 | 2.52 | 4.81 | 2.30 |
| NCEI | 3.30 | 4.04 | 3.68 | 3.95 |
| **SDUST(no HY-2A)** | 1.39 | 1.89 | 3.38 | 2.28 |

Final, SDUST2021GRA, DTU17 and SIO V30.1 are compared with shipborne data in the global area. The RMSs of corresponding differences are 4.42 mGal, 4.63 mGal, and 4.60 mGal. The STD of gravity crossover discrepancies of shipborne data in all domain is 5.27 mGal, so the accuracy of shipborne data is 3.73 mGal. Therefore, the accuracy of SDUST2021GRA, DTU17 and SIO V30.1 is 2.37 mGal, 2.74 mGal and 2.69 mGal, respectively.

Considering the accuracy of altimeter-derived gravity are efficiently affected by the coastline, the statistical information of differences between altimeter-derived models and shipborne data in different distances from the coastline is listed in Table 5.

**Table 5. Statistics of differences between altimetric and shipborne gravity in differences distances from coastline (Unit: mGal)**

| Distance from coastline (km) | Gravity model | Mean | STD | RMS | Precision of gravity model |
|---|---|---|---|---|---|
|  | SDUST | -0.90 | 8.11 | 8.16 | 7.20 |
| 0~10 | DTU | -1.84 | 9.12 | 9.31 | 8.32 |
|  | SIO | -0.56 | 11.00 | 11.00 | 10.35 |
|  | SDUST | -0.39 | 5.83 | 5.84 | 4.48 |
| 10~20 | DTU | -0.21 | 6.34 | 6.34 | 5.13 |
|  | SIO | 0.64 | 6.50 | 6.53 | 5.32 |
|  | SDUST | -0.12 | 5.03 | 5.04 | 3.37 |
| 20~30 | DTU | 0.23 | 5.44 | 5.44 | 3.96 |
|  | SIO | 0.42 | 5.22 | 5.24 | 3.65 |

| | | | | | |
|---|---|---|---|---|---|
| 30~40 | SDUST | 0.10 | 4.76 | 4.76 | 2.96 |
| | DTU | 0.02 | 5.09 | 5.09 | 3.46 |
| | SIO | 0.58 | 4.83 | 5.86 | 3.07 |
| 40~50 | SDUST | 0.10 | 4.57 | 4.57 | 2.64 |
| | DTU | 0.14 | 4.89 | 4.89 | 3.16 |
| | SIO | 0.63 | 4.59 | 4.63 | 2.67 |
| >50 | SDUST | 0.03 | 4.04 | 4.04 | 1.55 |
| | DTU | 0.08 | 4.14 | 4.14 | 1.80 |
| | SIO | 0.18 | 3.96 | 3.96 | 1.33 |

As the accuracy of shipborne data in all domain is 3.73 mGal and is rarely affected by the coastline, that of shipborne data can be considered of 3.73 mGal in different distances from the coastline. Thus, the accuracy of altimeter-derived models can be obtained from the differences in Table 5. The accuracy of SIO V30.1 is the highest and that of SDUST2021GRA is about 1.5 mGal, when the distance from the coastline is greater than 50 km. The accuracy of SDUST2021GRA is the highest within 50 km from the coastline, especially within 20 km.

In conclusion, the accuracy of SDUST2021GRA in all domain is 2.37 mGal, which is better than that of DTU17 and SIO V30.1, especially in offshore areas and areas with islands. There are three reasons for the high accuracy of SDUST2021GRA. First, HY-2A-measured altimeter data which are proved to have the important role in gravity anomaly recovery are used to derive gravity anomalies. Second, in areas between 40°S~40°N, XGM2019e up to degree and order 2159 is used as the reference gravity field model, which is from DTU13 over the oceans (Zingerle et al., 2020). The reference gravity field model of DTU17 and SIO V30.1 is EGM2008, which is from DNSC07 over the oceans (Pavlis et al., 2012). DTU13 is the successor model to DNSC07, and have the better accuracy and resolution (Andersen et al., 2014). Final, accurate L2p Version 3.0 products are used. Corrections (ancillary data and models) are updated and quality controls are performed for L2p products (CNES, 2020), making the high quality of L2p products. Moreover, the accuracy of SDUST2021GRA in the open oceans and areas with many islands is consistent with that of modern shipborne gravity.

## 4.3 Error information on grids of SDUST2021GRA

High-resolution error information of SDUST2021GRA is useful for potential users. Therefore, following the method proposed by Sandwell et al. (2021), first, for each mission of each satellite, the median absolute deviation of the along-track geoid gradients with respect to gridded DOVs in a block (10 min longitude and 6 min latitude) is calculated. The median is presumed related to the noise in the along-track geoid gradients. Then, the average of median for all missions of all satellites is divided that by the square root of the number of observations in every block. These values can be used to approximate the accuracy of gravity anomalies, because that accuracy of along-track geoid gradients is approximately proportional to that of

altimeter-derived gravity anomalies (Sandwell et al., 2013). Final, the overall map of approximate precision of SDUST2021GRA in Figure 10 is calibrated using a scaling factor that makes the value in Area A equal to the 1.39 mGal.

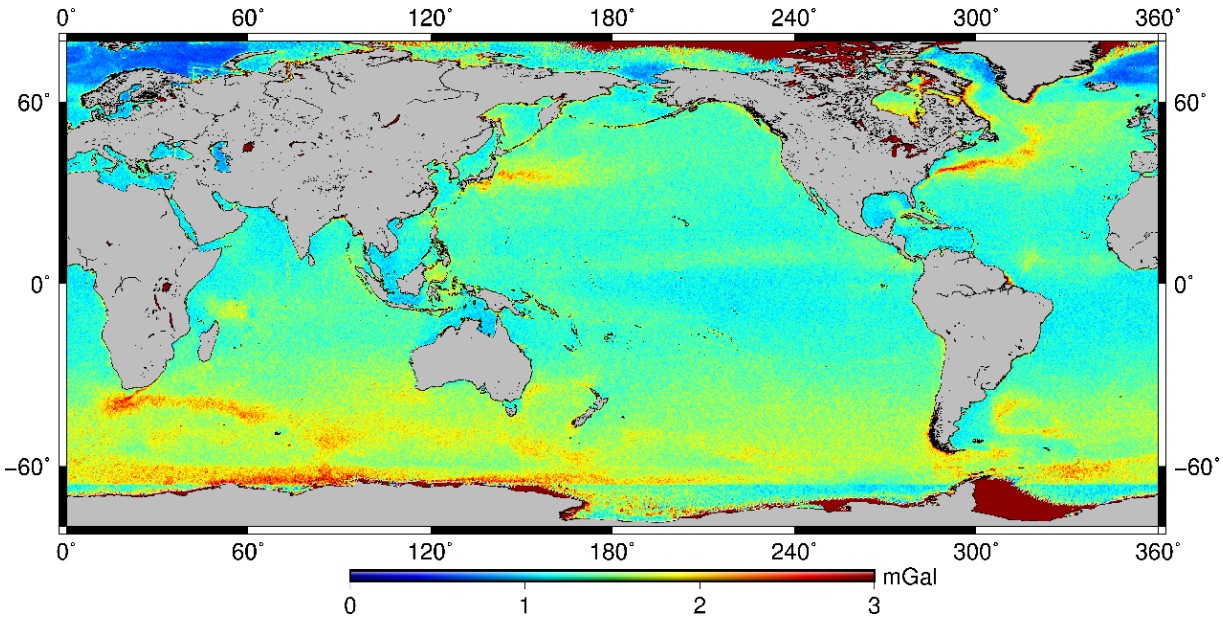

**Figure 10. Map of gravity anomaly error based on deviations of along-track geoid gradients from all altimeters.**

In order to compared with the results assessed by shipborne gravity, the accuracy of SDUST2021GRA in Area A, B, C and D is calculated by averaging values at the shipborne observation points by interpolation of the gridded errors in Figure 10, respectively. The corresponding accuracy is 1.39 mGal, 2.66 mGal, 3.47 mGal and 1.72 mGal. The accuracy of gravity in Area C is the lowest and that in Area A is the highest, which is the same as that evaluated using shipborne data. However, the accuracy in Area B is lower than that in Area D, which is different from that evaluated using shipborne data. This is because that Area D has the larger land area and more complex coastlines than those in Area B. Gravity anomaly in a grid point is derived from along-track geoid gradients in a large area around the point, so the land and coastlines have more effects on gravity anomalies than those on along-track geoid gradients. The accuracy of along-track geoid gradients can only be used to assess approximately that of altimeter-derived gravity anomalies.

**4.4 Data Availability**

The global marine gravity anomaly model (SDUST2021GRA) is available on the web site of https://doi.org/10.5281/zenodo.6668159 (last access: 27 June 2022) (Zhu et al., 2022). The dataset includes geospatial information (latitude, longitude) and free-air gravity anomalies.

## 5 Conclusions

During the processing of constructing the recognized global marine gravity anomaly models (DTU17 and SIO V30.1), HY-2A-measured altimeter data are not used, and Ka-band data are hardly specifically processed. Therefore, for improving the accuracy of gravity anomalies in offshore waters, multi-satellite altimeter data including HY-2A-measured SSHs are used to construct the global gravity anomaly model of SDUST2021GRA on 1′×1′ grids. In the processing, noise variance of Ka-band along-track geoid gradients in GM is determined by the different method from those of Ku-band observations. First, the SSH measurements are preprocessed, including Gaussian filtering for GM and simplified collinear adjustment for ERM. Second, along-track geoid gradients are calculated from preprocessed SSHs, and their accuracy is assessed by different methods, including the iteration method for Ka-band GM and crossover discrepancies of SSHs for other missions. Third, gridded DOVs are calculated by the LSC method based on the along-track geoid gradients and their accuracy. Final, SDUST2021GRA is derived from gridded DOVs by IVM. SDUST2021GRA is compared with DTU17 and SIO V30.1. Meanwhile, Shipborne gravity from NCEI is adjusted by the quadratic polynomial and used for assess the altimeter-derived gravity models.

The following conclusions can be drawn. The differences between SDUST2021GRA and DTU17 are slightly smaller than that when replacing DTU17 with SIO V30.1, and the differences between SDUST2021GRA and the two models are mainly small than 5 mGal. These indicates that SDUST2021GRA is reliable. Assessed by the shipborne gravity, the accuracy of SDUST2021GRA in the global is 2.37 mGal, which is better than that of DTU17 (2.74 mGal) and SIO V30.1 (2.69 mGal). In different distances from the coastline, the accuracy of SDUST2021GRA is more than 1 mGal higher than that of the two models within 10 km, and more than 0.6 mGal higher from 10 km to 20 km. HY-2A-measured altimeter data have an important role on gravity anomaly recovery in areas with complex coastlines and many islands. The accuracy of gravity anomalies derived from multi-satellite altimeter data without HY-2A in the areas is about 2.5% lower than that with HY-2A.

All these verifications show that SDUST2021GRA reaches an international advanced level of altimeter-derived gravity anomaly models. The accuracy of SDUST2021GRA is better than that of DTU17 and SIO V30.1 in the global area, especially in the offshore area and the area with many islands. Moreover, the accuracy of SDUST2021GRA is consistent with that of modern shipborne gravity in the open ocean and the area with islands, and better than that of NCEI shipborne gravity.

**Table A1. RMS of differences between altimeter-derived and shipborne gravity data (mGal)**

|     |       | B1    | B2   | B3   | B4   | B5   | B6    | B7    | B8   |
|-----|-------|-------|------|------|------|------|-------|-------|------|
|     | SDUST | 1.98  | 6.40 | 5.17 | 3.81 | 3.68 | 3.43  | 3.93  | 4.01 |
| L1  | DTU   | 1.90  | 6.33 | 5.26 | 3.88 | 3.85 | 3.59  | 4.04  | 4.07 |
|     | SIO   | 1.94  | 6.19 | 5.09 | 3.81 | 3.72 | 3.89  | 4.01  | 3.81 |
|     | SDUST | 7.93  | 7.60 | 5.25 | 8.88 | 6.00 | 6.18  | 7.77  | 2.82 |
| L2  | DTU   | 7.48  | 7.68 | 5.3  | 8.48 | 6.74 | 6.61  | 7.98  | 2.49 |
|     | SIO   | 10.96 | 7.78 | 5.39 | 9.35 | 5.88 | 8.18  | 7.98  | 2.8  |
|     | SDUST | 3.96  | 7.19 | 6.02 | 4.35 | 6.66 | 4.10  | 8.33  | 1.65 |
| L3  | DTU   | 3.96  | 7.10 | 7.16 | 4.37 | 7.37 | 4.21  | 10.21 | 1.43 |
|     | SIO   | 5.42  | 6.96 | 6.65 | 4.39 | 6.73 | 6.92  | 5.40  | 1.36 |
|     | SDUST | 2.76  | 4.19 | 4.82 | 3.04 | 3.31 | 3.16  | - *   | -    |
| L4  | DTU   | 2.73  | 4.19 | 4.79 | 3.07 | 3.34 | 3.23  | -     | -    |
|     | SIO   | 2.84  | 4.06 | 4.78 | 3.02 | 3.28 | 3.21  | -     | -    |
|     | SDUST | 2.78  | 3.84 | 3.10 | 3.42 | 4.71 | 11.90 | -     | -    |
| L5  | DTU   | 2.76  | 3.78 | 3.02 | 3.44 | 4.88 | 12.02 | -     | -    |
|     | SIO   | 2.97  | 3.66 | 2.92 | 3.35 | 4.83 | 11.85 | -     | -    |
|     | SDUST | 3.72  | 3.70 | 4.36 | 4.72 | 4.49 | 5.19  | -     | -    |
| L6  | DTU   | 3.78  | 3.58 | 4.06 | 4.93 | 4.70 | 5.24  | -     | -    |
|     | SIO   | 3.76  | 3.40 | 4.01 | 5.03 | 4.53 | 5.24  | -     | -    |
|     | SDUST | 3.18  | 4.12 | 4.61 | 5.50 | 5.96 | 5.08  | 4.11  | -    |
| L7  | DTU   | 3.17  | 4.06 | 4.47 | 6.30 | 6.79 | 5.48  | 4.25  | -    |
|     | SIO   | 3.25  | 3.80 | 4.47 | 6.35 | 6.43 | 5.34  | 4.05  | -    |
|     | SDUST | 4.25  | 5.94 | 4.44 | 4.96 | 4.37 | 4.44  | 6.37  | -    |
| L8  | DTU   | 4.32  | 5.99 | 4.42 | 5.55 | 4.60 | 4.46  | 6.44  | -    |
|     | SIO   | 5.23  | 5.99 | 4.84 | 5.15 | 4.30 | 4.40  | 6.82  | -    |
|     | SDUST | 8.26  | 4.69 | 4.16 | 4.96 | 3.91 | 4.56  | 4.54  | 2.36 |
| L9  | DTU   | 8.30  | 4.82 | 4.43 | 5.32 | 4.55 | 4.67  | 4.54  | 3.21 |
|     | SIO   | 10.74 | 4.70 | 4.39 | 5.56 | 3.83 | 4.39  | 4.47  | 1.88 |
|     | SDUST | 3.92  | 4.00 | 5.04 | 4.22 | 3.55 | 3.76  | 4.48  | 6.53 |
| L10 | DTU   | 4.13  | 3.91 | 5.20 | 4.56 | 3.63 | 3.91  | 4.52  | 6.68 |
|     | SIO   | 10.11 | 3.89 | 5.12 | 4.43 | 3.44 | 3.56  | 4.68  | 6.87 |
|     | SDUST | 9.15  | 4.74 | 2.70 | 3.41 | 4.15 | 5.38  | 4.50  | 5.82 |
| L11 | DTU   | 8.66  | 4.65 | 2.66 | 4.46 | 4.43 | 4.80  | 4.51  | 5.83 |
|     | SIO   | 7.53  | 4.57 | 2.47 | 4.58 | 4.55 | 5.38  | 4.65  | 6.60 |
|     | SDUST | 8.56  | 4.73 | 2.64 | 2.59 | 2.76 | 3.73  | 3.84  | -    |
| L12 | DTU   | 8.62  | 4.60 | 2.76 | 3.01 | 2.74 | 3.60  | 3.78  | -    |
|     | SIO   | 8.16  | 4.45 | 2.54 | 2.69 | 2.61 | 3.72  | 4.37  | -    |
|     | SDUST | 5.88  | 7.60 | 3.69 | 2.45 | 3.55 | 4.88  | -     | -    |
| L13 | DTU   | 5.94  | 7.57 | 3.84 | 2.57 | 3.84 | 5.47  | -     | -    |
|     | SIO   | 5.44  | 7.48 | 3.55 | 2.35 | 3.45 | 5.20  | -     | -    |
|     | SDUST | 3.43  | 4.61 | 2.86 | 3.76 | 3.45 | 3.13  | -     | 3.17 |
| L14 | DTU   | 3.42  | 4.26 | 2.94 | 4.58 | 3.81 | 3.18  | -     | 3.81 |
|     | SIO   | 3.86  | 4.08 | 2.46 | 4.86 | 3.50 | 3.40  | -     | 5.86 |
|     | SDUST | 7.24  | 8.28 | 3.54 | 4.11 | 5.27 | 4.70  | 3.08  | 4.68 |
| L15 | DTU   | 7.44  | 8.42 | 4.01 | 4.47 | 5.81 | 4.80  | 2.98  | 4.64 |
|     | SIO   | 7.16  | 8.70 | 3.95 | 4.08 | 6.48 | 5.69  | 3.27  | 6.04 |

| | | | | | | | | |
|---|---|---|---|---|---|---|---|---|
| | SDUST | 7.68 | 5.06 | 4.06 | 2.89 | 3.88 | 3.93 | 2.68 | 3.87 |
| L16 | DTU | 7.83 | 5.01 | 4.28 | 3.30 | 3.90 | 4.05 | 2.47 | 3.60 |
| | SIO | 10.11 | 4.99 | 4.19 | 2.96 | 3.55 | 3.71 | 3.13 | 4.19 |
| | SDUST | 5.36 | 5.84 | 4.74 | 3.76 | 4.85 | 6.05 | 3.64 | 2.77 |
| L17 | DTU | 5.39 | 6.05 | 4.72 | 3.94 | 4.90 | 6.18 | 3.58 | 2.83 |
| | SIO | 9.12 | 5.92 | 4.67 | 3.73 | 4.91 | 6.04 | 3.33 | 2.76 |
| | SDUST | 14.45 | 3.90 | 4.79 | 3.99 | 4.38 | 3.90 | 3.60 | 3.17 |
| L18 | DTU | 14.96 | 3.81 | 4.88 | 6.04 | 4.45 | 3.97 | 3.54 | 3.15 |
| | SIO | 12.02 | 3.49 | 4.55 | 3.49 | 4.26 | 3.89 | 3.47 | 2.89 |

\* The signal of – means that there is no shipborne data in the region.

**Author contributions.** All authors have contributed to designing the approach and writing the manuscript.

**Competing interests.** The contact author has declared that neither they nor their co-authors have any competing interests.

**Disclaimer.** Publisher's note: Copernicus Publications remains neutral with regard to jurisdictional claims in published maps and institutional affiliations.

**Acknowledgments.** We are very grateful to AVISO for providing the altimeter data, and NCEI for providing ship-borne gravity. We are also thankful to SIO and DTU for marine gravity anomaly models. Tanks to ICGEM for providing reference gravity field.

**Financial support.** This work was partially supported by the National Natural Science Foundation of China (grants 41774001), the SDUST Research Fund (grant 2014TDJH101), and the Autonomous and Controllable Project for Surveying and Mapping of China (grant No. 816-517).

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
