# Peer review of "SDUST2021GRA: Global marine gravity anomaly model recovered from Ka-band and Ku-band satellite altimeter data"

_Earth System Science Data, 2022_

## Referee Comment (RC1)

Review of the manuscript: SDUST2021GRA: Global marine gravity anomaly model recovered from Ka-band and Ku-band satellite altimeter data, by Zhu et al.

**Summary**
The paper focuses on the determination of a global marine gravity anomaly model based on multi-satellite altimetry data. The content of the manuscript is interesting. Although the method used is well-established in geodetic literature, the advantage of the paper is that the noise variances of along-track geoid gradients in Ka band and Ku band are determined by an iterative method and crossover discrepancies in the processing of gravity anomaly recovery. Meanwhile, HY-2A data are used to establish the global marine gravity model of SDUST2021GRA, unlike the recognized marine gravity models (DTU17 and SIO V30.1).

The resulting SDUST2021GRA model is validated with the existing marine gravity survey data, including the recognized marine gravity anomaly models and ship-borne gravity data. The validation results confirm that accuracy of SDUST2021GRA is comparable to contemporary marine gravity data by the applied method and the combination of multi-satellite altimetry data. Especially over the offshore area, SDUST2021GRA has higher accuracy. This model may beneficial for marine gravity field recovery and mean dynamic topography refinement as well as other researches like the study of ocean state and ocean circulation.

**Detailed comments**
Some minor comments are listed as below:

**2.1 Study area**

(1) Please add more information for the figure caption of Figure 1 on page 3, for instance, what are these colors represent? Although we know from the text that red/green displays XGM2019e/ EGM2008, and blue shows shipborne gravity data. However, it may be more clear by adding these captions, and readers can directly catch the meanings of this figure without finding materials from the text in different locations.

**3.2.3 Noise variances of Ka-band and Ku-band geoid gradients**

(2) The iterative method for determining the noise variances are presented by a long text. It is better to add a flowchart and shorten the text.

**4 Gravity anomaly results**

(3) As the authors stated in introduction on page 2 line 40 "HY-2A-measured altimeter data are rarely used for published global models of gravity anomalies", while the development of SDUST2021GRA included HY-2A data. To quantify the effects of HY-2A data on global gravity anomalies inversion, the authors may consider additionally computing one solution without HY-2A, the difference between this solution and SDUST2021GRA can study the contribution of HY-2A data. This model can also be validated against shipborne gravity data to assess its quality, and compared with SDUST2021GRA.

**4.1 Comparison with SIO V30.1 and DTU17**

(4) In this study, SDUST2021GRA was compared with DTU17 and SIO V30.1 for cross validation, and concluded that the former has better quality over offshore areas. It is noticeable that the SAR altimeter data from Sentinel-3A/B had been used in the computation of SIO V30.1, and it is well known that these SAR data has great advantages over traditional radar altimeter over coasts. While, these SAR altimeter data were not used in developing SDUST2021GRA (as shown in Table 1 on page 4 line 85), the authors may consider using SAR altimeter data to further improve the marine gravity anomaly model.

(5) Figure 5 shows the differences between SDUST2021GRA and the two recognized marine gravity anomaly models (DTU17 and SIO V30.1). The differences between DTU17 and SIO V30.1 should be also calculated to illustrate the differences of gravity anomaly models. Please also give the error information (the formal errors) of these three models if it is possible.

(6) There is no data in some areas in Figure 8 on page 15. Please explain it in more details in the manuscript.

(7) Please plot a global picture of formal error of SDUST2021GRA via the law of error propagation, as the same did by the authors on page 16, line 345. This picture contains the error information of SDUST2021GRA over different areas, which are useful for the potential users. Moreover, the authors may consider comparing the difference against shipborne gravity data of SDUST2021GRA and its formal errors over different areas, to see if the formal error of SDUST2021GRA is reliable.

---

## Author Comment (AC1)

Thank you for your comments and meticulous check of our manuscript that make the paper more interesting and informative. Below, please find our point to point responses to the comments in the list of revisions.

We have made a revision to the manuscript and displaced some materials in the text. We have marked all the revisions in red on the revised manuscript.

**A list of revisions**

1.  Please add more information for the figure caption of Figure 1 on page 3, for instance what are these colors represent? Although we know from the text that red/green displays XGM2019e/ EGM2008, and blue shows shipborne gravity data. However, it may be more clear by adding these captions, and readers can directly catch the meanings of this figure without finding materials from the text in different locations.

Answer: Thanks for the suggestion. The caption is rephrased:

**Figure 1. Division map for deriving gravity anomalies and tracks of NCEI shipborne data. The red and green areas means that the reference gravity anomalies are obtained from XGM2019e and EGM2008, respectively. The lines in blue present the cruises of shipborne gravity. The areas in cyan boxes are the special areas for analysis.**

2.  The iterative method for determining the noise variances are presented by a long text.
    It is better to add a flowchart and shorten the text.

Answer: The flowchart of the iterative method is added. The text is shortened.

[Figure]

**Figure 3. Iterative method of assessing the accuracy of SRL/DP-measured along-track geoid gradients. Initial accuracy of geoid gradients is only used for the first calculation of gravity, and new accuracy is used for other calculations.**

The accuracy of along-track geoid gradients of SRL/DP can be obtained following the method in Figure 3. First, initial precision of along-track geoid gradients of SRL/DP is also assessed by the Equation (8). Gravity anomalies are derived from SRL/DP-measured SSHs based on the initial precision. Second, the precision of SRL/DP-derived gravity is assessed by shipborne gravity and SIO V30.1 model, then used to calculate the new precision of along-track geoid gradient from

SRL/DP by Equation (9)(Equation(10) in the original manuscript). The new SRL/DP gravity anomalies are derived based on the new precision of along-track geoid gradients. Final, the calculation of step second is repeated, and terminated when the difference in precision of altimetric gravity between adjacent times is less than 0.02 mGal.

3. As the authors stated in introduction on page 2 line 40 "HY-2A-measured altimeter data are rarely used for published global models of gravity anomalies", while the development of SDUST2021GRA included HY-2A data. To quantify the effects of HY-2A data on global gravity anomalies inversion, the authors may consider additionally computing one solution without HY-2A, the difference between this solution and SDUST2021GRA can study the contribution of HY-2A data. This model can also be validated against shipborne gravity data to assess its quality, and compared with SDUST2021GRA.

Answer: Thanks for the good suggestion. The altimeter-derived gravity anomalies without HY-2A in Area A~D are obtained, and compared with shipborne gravity. The accuracy is added in Table4.

**Table 4. Accuracy of altimeter-derived and shipborne gravity data in different areas (Unit: mGal)**

|  | Area A | Area B | Area C | Area D |
|---|---|---|---|---|
| SDUST | 1.39 | 1.82 | 3.34 | 2.22 |
| DTU | 1.72 | 2.54 | 3.81 | 2.60 |
| SIO | 0.97 | 2.52 | 4.81 | 2.30 |
| NCEI | 3.30 | 4.04 | 3.68 | 3.95 |
| **SDUST(no HY-2A)** | 1.39 | 1.89 | 3.38 | 2.28 |

HY-2A-measured altimeter data are excluded from the multi-satellite altimeter dataset, and the residual altimeter data are used to derived gravity anomalies marked SDUST(no HY-2A), as listed in Table 4. Compared with the accuracy of SDUST2021GRA, that of SDUST(no HY-2A) reduces by 3.8%, 1.2% and 2.7% in Area B, C and D, respectively. These indicate that HY-2A has an important role in gravity anomaly recovery in areas with complex coastline and many islands.

4. In this study, SDUST2021GRA was compared with DTU17 and SIO V30.1 for cross validation, and concluded that the former has better quality over offshore areas. It is noticeable that the SAR altimeter data from Sentinel-3A/B had been used in the computation of SIO V30.1, and it is well known that these SAR data has great advantages over traditional radar altimeter over coasts. While, these SAR altimeter data were not used in developing SDUST2021GRA (as shown in Table 1 on page 4 line 85), the authors may consider using SAR altimeter data to further improve the marine gravity anomaly model.

Answer: Sentinel-3A/B perform the exact repeat mission (ERM) with the sparse track coverage. Data from ERM have a small effect on gravity recovery, so the data from Sentinel-3A/B are not used for SDUST2021GRA. That is a good suggestion. We will study on the performance of SAR altimeter data from Sentinel-3A/B on gravity recovery to further improve the marine gravity anomaly model in the future study.

5. Figure 5 shows the differences between SDUST2021GRA and the two recognized marine gravity anomaly models (DTU17 and SIO V30.1). The differences between DTU17 and SIO V30.1 should be also calculated to illustrate the differences of gravity anomaly models. Please also give the error information (the formal errors) of these three models if it is possible.

Answer: The differences between DTU17 and SIO V30.1 are calculated, and added as Figure 6(c).

The error information of DTU17 and SIO V30.1 cannot be obtained from their value and reference papers. The error information of SDUST2021GRA is added in the revised manuscript.

[Figure]

**Figure 6. Different between SDUST2021GRA and recognized marine gravity models: (a) is for SDUST2021GRA and SIO V30.1, (b) is for SDUST2021GRA and DTU 17, and (c) is for DTU 17 and SIO V30.1.**

High-resolution error information of SDUST2021GRA is useful for potential users. Therefore, following the method proposed by Sandwell et al. (2021), first, for each mission of each satellite, the median absolute deviation of the along-track geoid gradients with respect to gridded DOVs in a block (10 min longitude and 6 min latitude) is calculated. The median is presumed related to the noise in the along-track geoid gradients. Then, the average of median for all missions of all satellites is divided that by the square root of the number of observations in every block. These values can be used to approximate the accuracy of gravity anomalies, because that accuracy of along-track geoid gradients is approximately proportional to that of altimeter-derived gravity anomalies (Sandwell et

al., 2013). Final, the overall map of approximate precision of SDUST2021GRA in Figure 10 is calibrated using a scaling factor that makes the value in Area A equal to the 1.39 mGal.

[Figure]

**Figure 10. Map of gravity anomaly error based on deviations of along-track geoid gradients from all altimeters.**

6.   There is no data in some areas in Figure 8 on page 15. Please explain it in more details in the manuscript.

Answer: Thanks. The explanation is added in the manuscript.

In Figure 9 (Figure 8 in the original manuscript), there is no data in region L4B7, L5B7, L6B7 and L13B7, which is because that these areas have no sea. There is no data in other regions in Figure 9, which is caused by no shipborne gravity data in these areas.

7.   Please plot a global picture of formal error of SDUST2021GRA via the law of error propagation, as the same did by the authors on page 16, line 345. This picture contains the error information of SDUST2021GRA over different areas, which are useful for the potential users. Moreover, the authors may consider comparing the difference against shipborne gravity data of SDUST2021GRA and its formal errors over different areas, to see if the formal error of SDUST2021GRA is reliable.

Answer: That is a good suggestion.

High-resolution error information of SDUST2021GRA is useful for potential users. Therefore, following the method proposed by Sandwell et al. (2021), first, for each mission of each satellite, the median absolute deviation of the along-track geoid gradients with respect to gridded DOVs in a block (10 min longitude and 6 min latitude) is calculated. The median is presumed related to the noise in the along-track geoid gradients. Then, the average of median for all missions of all satellites is divided that by the square root of the number of observations in every block. These values can be used to approximate the accuracy of gravity anomalies, because that accuracy of along-track geoid gradients is approximately proportional to that of altimeter-derived gravity anomalies (Sandwell et al., 2013). Final, the overall map of approximate precision of SDUST2021GRA in Figure 10 is calibrated using a scaling factor that makes the value in Area A equal to the 1.39 mGal.

[Figure]

**Figure 10. Map of gravity anomaly error based on deviations of along-track geoid gradients from all altimeters.**

In order to compared with the results assessed by shipborne gravity, the accuracy of SDUST2021GRA in Area A, B, C and D is calculated by averaging values at the shipborne observation points by interpolation of the gridded errors in Figure 10, respectively. The corresponding accuracy is 1.39 mGal, 2.66 mGal, 3.47 mGal and 1.72 mGal. The accuracy of gravity in Area C is the lowest and that in Area A is the highest, which is the same as that evaluated using shipborne data. However, the accuracy in Area B is lower than that in Area D, which is different from that evaluated using shipborne data. This is because that Area D has the larger land area and more complex coastlines than those in Area B. Gravity anomaly in a grid point is derived from along-track geoid gradients in a large area around the point, so the land and coastlines have more effects on gravity anomalies than those on along-track geoid gradients. The accuracy of along-track geoid gradients can only be used to assess approximately that of altimeter-derived gravity anomalies.

---

## Author Comment (AC2)

Thank you for your comments and meticulous check of our manuscript that make the paper more interesting and informative. Below, please find our point to point responses to the comments in the list of revisions.

We have made a revision to the manuscript and displaced some materials in the text. We have marked all the revisions in red on the revised manuscript.

**A list of revisions**

**General comments**

1. The authors provide a well-documented dataset, also discussing its potential in different region of the global ocean. The methodology they pursued looks rigorous and the assessment of accuracy is solid. The authors, however, need to improve the writing of several parts (see specific comments). In particular, English grammar is very poor and the structure of several paragraphs looks often confusing.

Answer: We are sorry for the poor English. The revised manuscript is proofread by a English-speaking professor.

.

2. Finally, the authors should make a better effort in stressing the usefulness of the SDUST2021GRA, highlighting gaps of the existing products (in both introduction and conclusions).

Answer: The usefulness of the SDUST2021GRA is added in in both introduction and conclusions.

In introduction:

Accuracy of altimeter-derived gravity anomalies in offshore waters is low because of the waveform contamination by land. Compared with traditional Ku/C-band altimeters, Ka-band altimeter with higher frequency has a smaller altimeter footprint (CNES, 2016a), which leads to the smaller contamination radius of land. Moreover, the gravity anomaly model derived from more altimeter data is more accurate. For the recognized marine gravity anomaly models, HY-2A-measured altimeter data are not used, and Ka-band data are hardly specifically processed. Therefore, we will construct the global marine gravity anomaly model (SDUST2021GRA) on a $1'\times1'$ grid from multi-satellite altimeter data including HY-2A-measured data. In the processing, noise variance of Ka-band along-track geoid gradients is determined by the different method from those of Ku-band observations.

In conclusions:

During the processing of constructing the recognized global marine gravity anomaly models (DTU17 and SIO V30.1), HY-2A-measured altimeter data are not used, and Ka-band data are hardly specifically processed. Therefore, for improving the accuracy of gravity anomalies in offshore waters, multi-satellite altimeter data including HY-2A-measured SSHs are used to construct the global gravity anomaly model of SDUST2021GRA on $1'\times1'$ grids. In the processing, noise variance of Ka-band along-track geoid gradients in GM is determined by the different method from those of Ku-band observations.

**Specific and minor comments**

1. Lines 39-42: the HY-2A satellite mission (and the payload) should be better introduce in this paragraph.

Answer: Thanks. The introduction of HY-2A is added in the revised manuscript.

HY-2A, China's first ocean dynamical satellite, was launched on August 16, 2011. A microwave imager, a dual-frequency (Ku band and C band) radar altimeter and a Ku-band scatterometer on HY-

2A are used to obtain brightness temperature, monitor basic ocean elements (sea level, significant wave height and wind speed) and determine sea surface vector wind field. Radar altimeter on HY-2A has perform geodetic mission for about four years.

2. Line 48-49: rephrase as "GM in LSC is determined by the iteration method which is proposed by Zhu et al. (2020)"

Line 57: rephrase as "…limited memory of the computer (Figure 1)"

Line 58: rephrase as "…marked from L1 to L18; from 80°S to 80°N, regions are marked from B1 to B8"

Answer: The sentences are rephrased.

3. Figure 1: caption of this figure should be improved by providing all the necessary details, i.e., what the colours indicate for the three different latitude belts; the authors should also write here what is indicated in line 59 regarding the B and L markers; I would also suggest to indicate in caption that shipborn traks are reported in blue, etc.

Answer: That is a good suggestion. The caption of Figure 1 is rephrased.

**Figure 1. Division map for deriving gravity anomalies and tracks of NCEI shipborne data. The red and green areas means that the reference gravity anomalies are obtained from XGM2019e and EGM2008, respectively. The lines in blue present the cruises of shipborne gravity. The areas in cyan boxes are the special areas for analysis. From 0° to 360°E, regions are marked from L1 to L18; from 80°S to 80°N, regions are marked from B1 to B8.**

4. Line 63: I guess the authors can indicate from the beginning that they are using the version 3.0 of the L2P products "Non-time critical Level 2 Plus (L2P) Version 3.0 products…", without adding any additional sentence below.

Answer: Thanks. The sentence is rephrased as the suggestion.

5. Line 71: rephrase as "altimetry data is gradually improved through the years. ERS-1 was launched before 1990"

Answer: The sentences are rephrased.

6. Line 80: as for several sentences, there is no need to write "as listed", "as showed", etc. It's much simpler to write, e.g., "ERM is marked with an '_B' after the satellite name (Table 1)

Answer: Thanks. Some sentences are rephrase.

7. Line 83: I do not think here is necessary to re-write "The study area is divided into multiple regions." I would remove this sentence.

Answer: This sentence is deleted in the revised manuscript.

8. Section 2.3.1: This whole section is really hard to follow. Rather than presenting the EGM2008 and the XGM2019e model at the beginning, I would suggest to change the logical thread, starting from (and improving) the last paragraph and, in particular, from lines 104-106.

Answer: The section is rephrased.

Gravity anomalies on regular grids can be obtained by the calculation function of gravity field functionals on ellipsoidal grids provided by International Centre for Global Earth Models (ICGEM) (http://icgem.gfz-potsdam.de/calcgrid). Earth Gravitational Field Model 2008 (EGM2008) (Pavlis et al., 2012) and XGM2019e (Zingerle et al., 2020) are the recognized high-precision Earth Gravitational Field models.

EGM2008 is complete to degree and order 2159, with additional coefficients up to degree 2190 and order 2159. Its half-wavelength resolution is about 9 km. XGM2019e (Zingerle et al., 2020) is a combined

global gravity field model complete to degree and order 5399. Its half-wavelength resolution is about 4 km. XGM2019e is mainly constructed from the GOCO06s satellite-only gravity field model, 15′ ground gravity dataset provided by the US National Geospatial-Intelligence Agency and 1′ augmentation dataset.

Since ICGEM only provides the calculation for XGM2019e up to degree and order 2159 (marked as the XGM2019e_2159 model), XGM2019e up to degree and order 2159 is used as the reference gravity field model. Compared with shipborne gravity anomalies, the differences for XGM2019e are greater than those for EGM2008 in some sea areas of the middle to high latitudes. Therefore, EGM2008 up to the order 2160 is used as the reference gravity field model in the areas between 40°S~80°S or 40°N~80°N referring to previous studies (Sandwell et al., 2014a; Shih et al., 2015). As shown in Figure 1, the reference gravity field model of the regions in red is XGM2019e up to degree and order 2159, and that in green is EGM2008 up to the order 2160.

9. Line 108: Is this first sentence really useful? It actually does not add any info.

Answer: This sentence is deleted in the revised manuscript.

10. Line 109: is this SIO 30.01, according to Table 2 ?

Answer: We are sorry for the mistake. SIO V30.01 is rephrased as SIO V30.1.

11. Line 119: rephrase as "…different countries and institutions in different time; a feature that requires some processing".

Answer: Thanks. The sentence is rephrased as the suggestion.

12. Line 121: rephrase as "Geoid heights can be calculated from SSHs by subtracting the dynamic topography".
    Line 123: do the authors mean "MDT-CNES-CLS18 (Mulet et al., 2021) is the most updated MDT model released by AVISO" ?
    Line 133: rephrase as "Moreover, there are some long-wavelength errors in shipborne gravity".
    Line 134: please, correct "which are caused by drifts in gravimeter readings".
    Line 149: I would simply write, for instance, "are parameters obtained, for each cruise, by least-square fitting from Eq. (1)".
    Line 154: rephrase as "gravity anomalies can be obtained from".

Answer: Thanks. The sentences are rephrased as the suggestion.

13. Line 160: "Jason-2/GM altimeter data are from GDRs whose…" is already said in section 2.2 (no need to repeat here).

Answer: This sentence is rephrased as "The reference ellipsoid used for Jason-2/GM altimeter data is different from WGS84 used for L2P products."

14. Equation (2): I would suggest to place dα immediately after (or before) the fraction lines.

Answer: Thanks, the sentences " $\mathrm{d}a$ is the difference between semimajor axis of WGS84 ellipsoid and that of T/P ellipsoid. $\mathrm{d}\alpha$ is the difference between flattening of the two ellipsoids." is placed immediately after the fraction lines.

where, $\mathrm{d}a$ is the difference between semimajor axis of WGS84 ellipsoid and that of T/P ellipsoid, and $\mathrm{d}\alpha$ is the difference between flattening of the two ellipsoids. $a$ and $e$ are…

15. Line 169: rephrase as "…and that of T/P ellipsoid, i.e., dα is the difference between…".

Answer: In these sentences " $\mathrm{d}a$ is the difference between semimajor axis of WGS84 ellipsoid and that of T/P ellipsoid. $\mathrm{d}\alpha$ is the difference between flattening of the two ellipsoids", the two variables are $\mathrm{d}a$ and $\mathrm{d}\alpha$, so we think "i.e." is not suitable. They are rephrased as "…that of T/P ellipsoid, and $\mathrm{d}\alpha$ is the difference between…"

16. Figure 2: the authors should provide a proper caption for this figure (e.g., all symbols are missing; define dashed line, etc.).

Answer: The caption of the figure is rephrased as

**Figure 2. Simplified collinear adjustment method.** $R_i$ **is the observation point on the reference track.** $P_k$ **and** $P_{k+1}$ **are the observation points on the other track. The latitude of** $R_i'$ **is the same as that of** $R_i$ **. The dashed line indicates the parallel.**

17. Line 190: is there any reference for this sentence?

Answer: A reference is added. "…residual geoid heights (Sansò and Sideris, 2013)"

Sansò, F., Sideris, M.: Observables of physical geodesy and their analytical representation. In: Sansò, F., Sideris, M. (eds) Geoid Determination, Lecture Notes in Earth System Sciences, vol 110. Springer, Berlin, Heidelberg. https://doi.org/10.1007/978-3-540-74700-0_2, 2013.

18. Line 219: rephrase as "The noise variance of SSHs can be obtained by…".

Answer: Thanks. The sentence is rephrased as the suggestion.

19. Lines 238-272: as I indicated in the general comments, a scheme (i.e., a figure) that can summarize this whole processing chain would be helpful. Figure 3 is introduced later on and it is not exhaustive.

Answer: The flowchart of the iterative method is added. The text is shortened.

[Figure]

**Figure 3. Iterative method of assessing the accuracy of SRL/DP-measured along-track geoid gradients. Initial accuracy of along-track geoid gradients is only used for the first calculation of gravity, and new accuracy is used for other calculations.**

The accuracy of along-track geoid gradients of SRL/DP can be obtained following the method in Figure 3. First, initial precision of along-track geoid gradients of SRL/DP is also assessed by the Equation (8). Gravity anomalies are derived from SRL/DP-measured SSHs based on the initial

precision. Second, the precision of SRL/DP-derived gravity is assessed by shipborne gravity and SIO V30.1 model, then used to calculate the new precision of along-track geoid gradient from SRL/DP by Equation (9)(Equation(10) in the original manuscript). The new SRL/DP gravity anomalies are derived based on the new precision of along-track geoid gradients. Final, the calculation of step second is repeated, and terminated when the difference in precision of altimetric gravity between adjacent times is less than 0.02 mGal.

20. Lines 274: the IVM formula was already introduced in Line 47. Here I would rephrase the sentence as "Vening-Meinesz Formula can be used to determine DOVs from gravity anomalies, so The inverse of Vening-Meinesz formula is used…"

Answer: Thanks. The sentence is rephrased as the suggestion.

21. Line 306 (and Figure 5): the authors should indicate in Figure 5 all geographical names that are mentioned in the text, i.e., the Aleutian Islands and the Philippon Islands. I would also indicate in Figure 5 the area that will be discussed in Figure 6, i.e., the South Sandwich Trench.

Answer: The geographical names are added in Figure 6(a) (Figure 5(a) in the original manuscript).

[Figure]

**Figure 6. Different between SDUST2021GRA and recognized marine gravity models: (a) is for SDUST2021GRA and SIO V30.1, (b) is for SDUST2021GRA and DTU 17, and (c) is for DTU 17 and SIO V30.1.**

22. Line 307: I guess this is a "Moreover" rather than a "Meanwhile". In the same line, what do the authors mean with "special submarine topography"? Which kind of characteristics they want to highlight?

Answer: "Meanwhile" is replaced by a "Moreover". The "special submarine topography" means the rapidly changing submarine topography. The sentence is rephrased as "Moreover, the differences in areas with rapidly changing submarine topography…"

23. Line 325: I would rename this title as "4.2 Shipborne gravity data assessment".

Answer: Thanks. The title is rephrased as the suggestion.

24. Figure 8: to make the reading of this figure straightforward, I would indicate in each panel the corresponding B region (i.e., B1 for panel (a), B2 for panel (b), and so on..).

Answer: Thanks. The figure is rephrased as the suggestion.

[Figure]

25. Line 332-339: this whole reasoning regarding L regions along B2 is not clearly coming out from Figure 8b (unless I am missing something). The authors should help the reader in visualizing from the data this interesting analysis.

Answer: In order to further compare the accuracy of each model, The differences between RMSs for the three models in Figure 9(b) in each region are calculated (Figure 10(a)). If the difference is greater than 0, it means that the accuracy of the former model is lower than that of the latter model. Therefore, In the 18 regions marked B2, the accuracy of SIO V30.1 are higher than that of SDUST2021GRA in 13 regions, and higher than that of DTU17 in 15 regions. Moreover, the differences between RMSs for the three models in 8 regions marked L15 (region L15 in Figure 9(a)~9(h)) are shown in Figure 10(b). The accuracy of SDUST2021GRA is higher than that of SIO V30.1 in 6 regions, and higher than that for DTU17 in 6 regions.

[Figure]

**Figure 10. Differences between RMSs. The RMSs are statistics of differences between altimeter-derived gravity models and shipborne data: (a) for region L1B2, L2B2,…, L18B2; (b) for region L15B1, L15B2,…, L15B8.**

26. Line 341: rephrase as "Second, four typical ocean areas, marked as A – D, are selected for analyzing"

Answer: Thanks. The sentence is rephrased as the suggestion.

27. Lines 350-352: here it is not clear what the authors mean with "open ocean" and "offshore" areas. Does "offshore" mean off coastal areas? The authors should indicate this in a clear way. Moreover, the authors should also provide some discussion regarding why "SDUST2021GRA has the best accuracy in the offshore areas and the areas with many islands" (see general comment)."

Answer: We are sorry for the unclear statement. The sentence is rephrased as "It can be seen from Figure 1 that the regions marked B2 are mainly the open sea areas and the regions marked L15 have complex coastlines."

The discussion is added.

In conclusion, the accuracy of SDUST2021GRA in all domain is 2.37 mGal, which is better than that of DTU17 and SIO V30.1, especially in offshore areas and areas with islands. There are three reasons for the high accuracy of SDUST2021GRA. First, HY-2A-measured altimeter data which are proved to have the important role in gravity anomaly recovery are used to derive gravity anomalies. Second, in areas between 40°S~40°N, XGM2019e up to degree and order 2159 is used as the reference gravity field model, which is from DTU13 over the oceans (Zingerle et al., 2020). The reference gravity field model of DTU17 and SIO V30.1 is EGM2008, which is from DNSC07 over the oceans (Pavlis et al., 2012). DTU13 is the successor model to DNSC07, and have the better accuracy and resolution (Andersen et al., 2014). Final, accurate L2p Version 3.0 products are used. Corrections (ancillary data and models) are updated and quality controls are performed for L2p products (CNES, 2020), making the high quality of L2p products.

28. Line 359: what do the authors mean with "global"? Do you mean "global area" or "all domain"? lease, rephrase.

Line 371: same as above.

Answer: Thanks. "in the global" is rephrased as "in all domain"

29. Line 380: rephrase as "SSHs are used to construct".

Line 397: same as for lines 371 and 359.

Answer: Thanks. The sentences are rephrased as the suggestion.30.

30.  Table A1: insert between the different L regions.

Answer: Thanks. Border lines are inserted between the different L regions.